# Development of Ex Vivo Analysis for Examining Cell Composition, Immunological Landscape, Tumor and Immune Related Markers in Non-Small-Cell Lung Cancer

**DOI:** 10.3390/cancers16162886

**Published:** 2024-08-20

**Authors:** Elena G. Ufimtseva, Margarita S. Gileva, Ruslan V. Kostenko, Vadim V. Kozlov, Lyudmila F. Gulyaeva

**Affiliations:** 1Federal Research Center of Fundamental and Translational Medicine, 2 Timakova Street, 630060 Novosibirsk, Russia; lfgulyaeva@gmail.com; 2V. Zelman Institute for the Medicine and Psychology, Novosibirsk State University, 1 Pirogova Street, 630090 Novosibirsk, Russia; gileva.rita@gmail.com; 3Novosibirsk Regional Clinical Oncology Dispensary, 2 Plakhotny Street, 630108 Novosibirsk, Russia; kostenko.rv@mail.ru (R.V.K.); vadimkozlov80@mail.ru (V.V.K.); 4Faculty of General Medicine, Novosibirsk State Medical University, 52 Krasny Prospect, 630091 Novosibirsk, Russia

**Keywords:** lung cancer, adenocarcinoma, squamous cell carcinoma, biomarkers, PD-L1, tumor microenvironment, tobacco smoking, macrophages, eosinophils

## Abstract

**Simple Summary:**

Lung cancer, including non-small-cell lung cancer (NSCLC), is one of the leading causes of cancer-related deaths worldwide. NSCLC treatment has undergone a major paradigm shift with the advent of targeted therapy and immunotherapy, which require the specific tumor and immune signatures of NSCLC to be tested to ensure careful patient selection and prediction of clinical benefits from the selected therapy. Tumor samples with large fibrotic areas and the focal nature of biomarker expression may be a source of bias in histological assays. The ex vivo analysis developed in this work allows us to rapidly explore various characteristics of cancer and tumor-infiltrating immune cells after their separation from fibrotic tissue in solid NSCLC samples and to have a better understanding of the immunological landscape and expression of different cell markers in the tumor as a whole, thereby avoiding some false negatives in the histological assay of the same tumor samples.

**Abstract:**

NSCLC is a very aggressive solid tumor, with a poor prognosis due to post-surgical recurrence. Analysis of the specific tumor and immune signatures of NSCLC samples is a critical step in prognostic evaluation and management decisions for patients after surgery. Routine histological assays have some limitations. Therefore, new diagnostic tools with the capability to quickly recognize NSCLC subtypes and correctly identify various markers are needed. We developed a technique for ex vivo isolation of cancer and immune cells from surgical tumor and lung tissue samples of patients with NSCLC (adenocarcinomas and squamous cell carcinomas) and their examination on ex vivo cell preparations and, parallelly, on histological sections after Romanovsky–Giemsa and immunofluorescent/immunochemical staining for cancer-specific and immune-related markers. As a result, PD-L1 expression was detected for some patients only by ex vivo analysis. Immune cell profiling in the tumor microenvironment revealed significant differences in the immunological landscapes between the patients’ tumors, with smokers’ macrophages with simultaneous expression of pro- and anti-inflammatory cytokines, neutrophils, and eosinophils being the dominant populations. The proposed ex vivo analysis may be used as an additional diagnostic tool for quick examination of cancer and immune cells in whole tumor samples and to avoid false negatives in histological assays.

## 1. Introduction

Non-small-cell lung cancer (NSCLC), which accounts for 85% of all lung cancers, is one of the most aggressive malignant tumors and the most common cause of cancer-related mortality worldwide [1,2], with tobacco smoking being the major risk factor for this disease [3,4]. Within the group of NSCLC, adenocarcinoma and squamous cell carcinoma are the two major histological subtypes exhibiting distinct epidemiologic characteristics, biological behavior, genetic alterations, clinical courses, prognoses, and treatment modality [5,6]. The capability to distinguish between these subtypes is of importance to improve management decisions for NSCLC patients [7]. While each of the majority of tumor specimens can be subtyped alone by examination of typical morphological features on common hematoxylin/eosin-stained histological sections [7], identification of immunohistochemical markers, mainly the combinations of TTF1 and CK7 for lung adenocarcinoma and p40/p63 and CK5/6 for lung squamous cell carcinoma, is also necessary to provide an accurate pathological diagnosis, especially a precise subclassification of poorly differentiated NSCLC with a high degree of cell pleomorphism, for personalized medicine [7,8,9,10].

Until recently, the treatment options of NSCLC included surgical resection (mostly for patients with early-stage and locally advanced disease), cytotoxic chemotherapy, radiation therapy, and combined modality chemoradiotherapy approaches; however, the benefits were limited [1,2]. Over the past decades, NSCLC management has evolved from conventional therapy to personalized targeted therapy using small-molecule inhibitors and antibodies for tumors harboring certain genetic aberrations in oncogenic drivers, such as *EGFR*, *ALK*, *ROS1*, *BRAF* and other genes, in cancer cells, mainly in lung adenocarcinomas [11,12]. In contrast, lung squamous cell carcinomas rarely harbor these driver mutations and/or gene translocations. Also, the occurrence of resistance mechanisms, such as novel mutations in these and other genes and the activation of alternative signaling pathways are observed in a large number of NSCLC patients with initial positive responses to these targeted therapies [13].

Today, the advent of immunotherapy with immune checkpoint inhibitors targeting the PD-1 receptor and its PD-L1 ligand has become a great help for NSCLC patients, particularly in the absence of associated driver mutations [14,15]. The immune checkpoint PD-L1 is a key immunoregulatory molecule, which, upon interacting with its receptor, PD-1, leads to immunosuppression so that the immune system cannot effectively surveil and eradicate cancer cells any further [11,15]. In NSCLC, PD-L1 is often overexpressed in cancer cells, thereby damping effector T-cell functions, and allows them to evade detection by the patients’ immune system [16,17,18]. NSCLC patients with tumor PD-L1 expression are considered to be the ones who benefit the most from immune checkpoint blockade [17,18,19,20]. Thus, PD-L1 protein detection by immunohistochemical examination of tumor specimens and evaluation of the PD-L1 tumor proportion score (TPS), which measures the percentage of marker-expressing cancer cells, are widely used as a predictive biomarker assay for anti-PD-1/PD-L1 therapies for NSCLC patients in routine clinical practice [17,19,20]. Nevertheless, some NSCLC patients with high levels of tumor PD-L1 have poor responses and, vice versa, patients with low or even negative PD-L1 expression exhibit good and durable responses to checkpoint inhibitors [18,19,20]. Such a discrepancy in the risk–benefit assessments of immunotherapy in NSCLC may be caused by multiple factors, including, on the one hand, a significantly heterogeneous, plastic, and dynamic expression of this marker in tumors [21,22,23,24,25,26,27,28], the availability of multiple clones of anti-PD-L1 antibodies [29,30,31], a variability between pathologists’ interpretation (especially for low PD-L1 TPS) [32,33], and sample quality issues [34] in diagnostic testing, and, on the other hand, the intrinsic and as-yet undiscovered biological complexity underpinning tumor-immunity interplay [35,36,37].

NSCLC and other solid tumors are composed not only of cancer cells, but also of heterogeneous subsets of non-transformed stromal cells, mainly cancer-associated fibroblasts and tumor-infiltrating immune cells from the myeloid and lymphoid lineage, establishing a complex tumor microenvironment (TME) with peculiar structural, biophysical, and secretory characteristics which shape tumor progression and might play a key role in the responses to therapies [35,36,37,38,39]. At the tumor–stroma interface, the immune activation response in cancer cells is believed to be likely to lead to the preferential expression of PD-L1 in this region [35]. Additionally, high tumor-infiltrating T lymphocyte density associates directly with immunotherapy benefit in NSCLC patients and predicts clinical outcomes [35,36,37,38]. Therefore, deciphering the immune cell landscape and specific immune signatures of NSCLC is critical for improving the efficacy of the currently used immune checkpoint inhibitors [40,41,42] and designing next-generation immunomodulatory strategies addressing the immune microenvironment within tumors [43,44,45].

Furthermore, PD-L1 expression can be regulated by the aryl hydrocarbon receptor (AhR), a transcription factor activated after binding to benzo[*a*]pyrene, one of the main components of tobacco smoke and the most studied carcinogen in it [46]. There have been clinical observations suggesting that NSCLC patients who smoke tobacco respond better to immunotherapy than their non-smoking peers [47]. After activation, AhR initiates the transcription of cytochrome P450 complex genes, including cytochrome P450 family 1 subfamily A member 1 (*CYP1A1*), the product of which plays an important role in the initiation and enhancement of malignant cell transformation through the formation of highly reactive compounds by oxidizing polycyclic aromatic hydrocarbons from tobacco smoke [46]. AhR activity is controlled by the disruption of the AhR-based transcriptional complex by the AhR repressor (AhRR), which leads to complete repression of the AhR-dependent target genes [48]. Data about the DNA methylation of the *AhRR* gene have recently been shown to provide eligibility criteria for screening those individuals, who are likely to develop lung cancer [49]. Understanding the role of the AhR/AhRR/CYP1A1 axis in lung cancer progression is of importance to improve current therapeutic approaches and develop new therapeutic strategies for the treatment of NSCLC.

Consequently, analysis of the specific tumor and immune signatures of NSCLC is a critical step in prognostic evaluation and management decisions. However, the commonly used assays with conventional and immunohistochemical staining of histological preparations after surgical resection of NSCLC have some limitations, such as a long test duration (3–5 days), difficulties in interpreting the results, and limited sample volumes available for test, because histological sections are normally 4–5 microns in width each and may not be representative for the entire tumor. Also, tumor samples with very large fibrotic areas and the focal nature of marker expression and immune cell composition may yield biased information regarding markers, cell compositions and immunological landscapes. Therefore, new diagnostic tools with the capability to quickly recognize NSCLC subtypes and correctly identify lung cancer-specific and immune-related markers are needed.

With this in mind and with reliance on our previous work on the isolation of cells, mainly alveolar macrophages, from lung lesions surgically removed from patients with pulmonary tuberculosis [50], we have developed a technique for separating cancer and tumor-infiltrating immune cells from fibrotic tissue in the tumor stroma of NSCLC and their rapid characterization on ex vivo cell preparations. In this study, we assessed tumor- and immune-related markers and found a close identity between the ex vivo results and the histological data from the same NSCLC tumor samples handled at once. This ex vivo analysis allowed us to make a more precise examination of PD-L1 expression and thus prevent false negatives from immunohistological findings on the same NSCLC samples.

## 2. Materials and Methods

### 2.1. Patients, Their Tumor and Lung Tissue Samples

Tumor and attendant lung tissue samples were obtained from 12 NSCLC patients, who underwent lung resection surgery (lobectomy or pneumonectomy) in the Department of Thoracic Surgery of the Novosibirsk Regional Clinical Oncology Dispensary (Novosibirsk, Russia) between November 2021 and April 2023. The patients’ clinical information, including age, gender, smoking status (from self-administered questionnaires indicating the age of smoking initiation and, for former smokers, the age of smoking cessation), histological type of tumor, such as adenocarcinoma (ad) and squamous cell carcinoma (sq), stage of lung cancer, EGFR mutations, and attendant pulmonology and immune diseases, was collected from medical records. The pathological diagnosis was determined using the 8th edition staging system. None of the patients had received neoadjuvant therapy. This study was approved by the Ethical Committee of the Novosibirsk Regional Clinical Oncology Dispensary (15/2021/11/16) and was conducted in accordance with the principles of the Declaration of Helsinki and its later amendments. Written informed consent was obtained from all patients. Immediately after surgery, tumor samples of randomly selected areas of tumor and, in parallel, lung tissue obtained from the lung part about 5 cm away from the tumor were collected for each patient and each sample was cut into two. One portion of the pieces was collected for ex vivo production of the patients’ cells; the other portion of the pieces was simultaneously submitted to histological examination.

### 2.2. Ex Vivo Isolation of Cells and Production of Ex Vivo Cell Preparations

Cells were isolated ex vivo from the samples of surgically resected tumor and attendant lung tissues with reliance on a technique that we had previously used to produce ex vivo cell cultures, mainly those of alveolar macrophages, from lung lesions surgically removed from patients with pulmonary tuberculosis [50]. In this work, samples were cut into small pieces and, to separate cell suspension containing cancer and immune cells from fibrotic tissue, were further rubbed through a metal screen of a sieve with pores 0.5–1.0 mm in diameter in phosphate-buffered saline (PBS, pH 7.4) without any enzymatic treatments. Cell pellets were centrifuged at 400× *g* for 5 min at room temperature. Precipitates were diluted in PBS and 100 µL of cell suspension was introduced into a single cytofunnel sample chamber (Tharmac Cellspin, Limburg an der Lahn, Germany), which was linked to a glass single cytoslide (Tharmac Cellspin, Limburg an der Lahn, Germany) for each ex vivo cell preparation obtained. Then, thin-layer cell preparations were made using the cytocentrifuge Awel MF20 (Domel, Železniki, Slovenia) run at 400× *g* for 5 min at room temperature. The ex vivo cell preparations were immediately fixed with 4% formaldehyde solution in PBS for 10 min at room temperature. Six ex vivo cell preparations were made by the same method for each tumor sample and, in parallel, for each attendant lung tissue sample. 

### 2.3. Cell Staining

To visualize cancer cells and different types of immune cells, some ex vivo cell preparations were washed with PBS and stained with a mixture of azure, eosin, and methylene blue in Romanovsky–Giemsa stain (Minimed, Moscow, Russia) according to the manufacturer’s instruction. 

After washing with PBS, the other ex vivo cell preparations were permeabilized with 0.3% Triton X-100 solution in PBS for 2 min, blocked in PBS solution containing 2% BSA, and incubated with the appropriate rabbit or mouse primary antibodies to human cancer cell markers: TTF1 (clone SP141, Abcam, Cambridge, UK, ab227652), p40 (clone EPR17863-47, Abcam, Cambridge, UK, ab203826), CK7 (clone PBM-12F1, PrimeBioMed, Moscow, Russia), PanCK, including the cytokeratin 5/6 (clone C11, PrimeBioMed, Moscow, Russia), Ki-67 (clone SP6, Thermo Fisher Scientific, Waltham, MA, USA, MA5-14520), AhR (clone RPT9, Abcam, Cambridge, UK, ab2769), AhRR (Abcam, Cambridge, UK, ab234817), CYP1A1 (MyBioSource, San Diego, CA, USA, MBS178240); to immune-related markers: PD-L1 (clone ABM4E54, Abcam, Cambridge, UK, ab210931), PD-1 (clone CAL20, Abcam, Cambridge, UK, ab237728), CD14 (clone SP192, Spring Bioscience, Pleasanton, CA, USA, M492), IFNγ (clone 25718, Thermo Fisher Scientific, Waltham, MA, USA, MA5-23718), TNFα (Thermo Fisher Scientific, Waltham, MA, USA, P300A), FGFb (clone 6/basic FGF, BD Biosciences, Franklin Lakes, NJ, USA, 610072), IL-1β (clone 2805, Thermo Fisher Scientific, Waltham, MA, USA, MA5-23691), IL-4 (Thermo Fisher Scientific, Waltham, MA, USA, PA5-25165), IL-10 (clone 945A2A5, Invitrogen, Waltham, MA, USA, AHC9102), IL-12 (clone 24945, Thermo Fisher Scientific, Waltham, MA, USA, MA5-23715), and to fibroblast markers: CD10 (clone GM-003, PrimeBioMed, Moscow, Russia), a membrane metalloendopeptidase [51], and TRITC-labeled phalloidin dye (Sigma-Aldrich, Burlington, MA, USA, P1951) to stain filamentous actin (F-actin), diluted 1:100 each. 

For all ex vivo cell preparations, fluorescent visualization of bound primary antibodies was achieved using goat polyclonal secondary antibodies to rabbit IgG conjugated with DyLight 488 or DyLight 594 (Thermo Fisher Scientific, Waltham, MA, USA, 35553 or 35561, respectively) and to mouse IgG conjugated with Alexa 488 (Thermo Fisher Scientific, Waltham, MA, USA, A-11001) diluted 1:400 each. The ex vivo cell preparations were incubated with the appropriate antibodies for 60 min at room temperature. Fluorescent staining was analyzed using the ProLong Gold Antifade Mountant with DAPI (Thermo Fisher Scientific, Waltham, MA, USA, P36935). Actively growing human larynx carcinoma Hep-2 cells (courtesy of PhD. M.V. Solomatina of the Federal Research Center of Fundamental and Translational Medicine, Novosibirsk, Russia) were used as the internal control to validate the adequacy of the Ki-67 staining reaction and were stained with the appropriate antibodies and Alexa 488-labeled phalloidin dye (Thermo Fisher Scientific, Waltham, MA, USA, A12379) in parallel with the NSCLC preparations.

For some ex vivo preparations, biochemical visualization of bound primary antibodies was achieved using the Hydrogen Peroxidase Blocking Reagent with 3% H_2_O_2_ solution and the Prime Vision kit with horseradish peroxidase (HRP) and diaminobenzidine (DAB) (PrimeBioMed, Moscow, Russia) as a substrate according to the manufacturer’s instructions. After DAB staining, the cells were further counterstained with Romanovsky–Giemsa stain, as described above.

### 2.4. Histology

For all patients, the histological sections of the resected tumor and attendant lung tissues were prepared in parallel with ex vivo cell expansion. In brief, each resected specimen was cut into two. One portion of the pieces was collected for making ex vivo cell preparations as described above. The other portion was fixed with 4% formaldehyde solution in PBS for 15 h at +4 °C. After fixation, the pieces of tumor and attendant lung tissue were washed with PBS, incubated with 30% sucrose in PBS (pH 7.4) for 20 h at +4 °C, frozen in Tissue-Tek O.C.T. Compound (Sakura Finetek, Torrance, CA, USA, 4583) at −25 °C, and sectioned into 16 µm slices using a Microtome Cryostat HM550 (Microm, Walldorf, Germany) at the Shared Center for Microscopic Analysis of Biological Objects of the Institute of Cytology and Genetics, SB RAS (Novosibirsk, Russia). Sections were air-dried on SuperFrost Plus slides (Thermo Fisher Scientific, Waltham, MA, USA) and stained with Romanovsky–Giemsa stain or antibodies as described above in parallel with the ex vivo cell preparations.

All histological sections (~4 × 4 mm each) were permeabilized with 0.3% Triton X-100 solution in PBS for 45 min, blocked, and incubated with the appropriate primary antibodies for 20 h at +4 °C and with the appropriate secondary antibodies for 60 min at room temperature. Fluorescent staining was analyzed using the ProLong Gold Antifade Mountant with DAPI as described above.

For some histological sections, biochemical visualization of bound primary antibodies was achieved using the Hydrogen Peroxidase Blocking Reagent and the Prime Vision kit as described above. After DAB staining, the cells were further counterstained with Romanovsky–Giemsa stain as described above.

### 2.5. Microscopy

The ex vivo cell and histological preparations were examined at the Shared Center for Microscopic Analysis of Biological Objects of the Institute of Cytology and Genetics, SB RAS (Novosibirsk, Russia), using an Axioskop 2 plus microscope (Zeiss, Oberkochen, Germany) and objectives with various magnifications (Zeiss, Oberkochen, Germany), and photographed using an AxioCam HRc camera (Zeiss, Oberkochen, Germany); the images were analyzed using the AxioVision 4.7 microscopy software (Zeiss, Oberkochen, Germany). All preparations stained with fluorescent dyes were examined under an LSM 780 laser scanning confocal microscope (Zeiss, Oberkochen, Germany) using the ZEN 2010 software (Zeiss, Oberkochen, Germany). All ex vivo cell preparations were analyzed for cell composition and the expression of different markers for each cell type. The cancer and immune cells were counted separately on each ex vivo cell preparation for each patient in each test. All cancer cells (from 800 to 2000 cells) were analyzed in each ex vivo cell preparation. For the histological preparations, three serial tumor and attendant lung tissue sections were analyzed for each staining in each test.

### 2.6. Statistical Analysis

Statistical data processing was performed using Prism 6.0 (GraphPad Software, La Jolla, CA, USA) and Microsoft Excel 2010. Statistical significance for the comparisons between the datasets was determined using Student’s *t*-test. Differences were considered statistically significant at *p* < 0.05.

## 3. Results

### 3.1. Clinicopathological Characteristics of NSCLC Patients

The patients’ characteristics regarding sex, age, smoking status, surgical procedure with the anatomic location of the resected materials, pathologic tumor stage, and attendant pulmonology diseases are presented in Table 1 according to the tumor’s histological NSCLC subtype and the patients’ tumor samples used in ex vivo analysis. The histological classification of the tumor samples was as follows: seven non-mucinous adenocarcinomas (tumor samples ad1–ad6 and ad8), one mucinous adenocarcinoma (tumor sample ad7), and four squamous cell carcinomas (tumor samples sq1–sq4). According to the medical records, EGFR mutation-positive tumor was found only in one patient with lung adenocarcinoma (tumor sample ad2), whose tumor had a classical oncogenic mutation in the form of the exon 19 deletion in the *EGFR* gene in the cancer cells. None of the patients had received neoadjuvant treatment. At the time of diagnosis, 9 out of 12 patients were current smokers with a long history of tobacco smoking (tumor samples ad1–ad5, ad8, sq1, sq2, and sq4), one patient was a former smoker (tumor sample sq3), one patient had never smoked (tumor sample ad7), and one patient was a non-smoker without any comments (tumor sample ad6).

### 3.2. Experimental Design and Cell Composition after Ex Vivo Isolation from Tumor Samples

All tumors used in our study were solid carcinomas (Figure 1a) and were assumed to have a large amount of fibrotic tissue in them. For each tumor sample, the fibrotic tissue was separated from the cell suspension by some homogenization of tumor samples, pre-cutting these into smaller pieces and washing off cancer cells using a sieve (Figure 1b). From the pellets obtained by centrifugation of the cell suspension, ex vivo cell preparations were made and analyzed after Romanovsky–Giemsa and immunofluorescent/immunochemical staining with NSCLC subtype-specific antibodies to different markers. As a result, mainly cancer cells were detected on the ex vivo cell preparations (Figure 1d,f), while the CD10- and F-actin-positive cancer-associated fibroblasts (Figure 1c,e) were identified mostly on the smears obtained from the stroma fibers isolated in the sieves for all tumor samples tested.

### 3.3. Tumor-Related Markers Expressed by the Cancer Cells That Were Studied in Ex Vivo and Histological Analyses at Once

The specific cellular features and, in some cases, the spatial architectures of the cancer cells and their clusters were detected not only by histological examination, but also by ex vivo analysis of the Romanovsky–Giemsa-stained preparations obtained from the same tumor samples—this is especially important for the characterization of lung adenocarcinomas (Figure 2), which were histologically heterogeneous and had significantly different prognoses. For example, the presence of poorly differentiated cancer cells with a high degree of cellular pleomorphism and a large cell size both on the ex vivo cell preparations and, in parallel, on the histological sections from tumor sample ad2 suggested that the tumor was behaving in a more aggressive fashion. Also, we obtained the NSCLC subtype-based pathomorphological characteristics of the tumor samples from the ex vivo and histological analyses, and found them identical to those in the medical records. The presence of extensive areas of cancer-associated fibroblasts was confirmed by histological analysis of the tumor samples. Of note, fibrotic tissue occupied up to 80–90% of the area of some histological sections, especially in tumor samples ad3 and sq1–sq3.

Detection of tumor-specific markers revealed similar numbers of cancer cells that were positive for lung adenocarcinoma-specific (TTF1 and CK7) or squamous-cell-carcinoma-specific (p40 and PanCK), carcinogenesis-related (AhR, AhRR, and CYP1A1 for tobacco smoking), and proliferation activity (Ki-67) markers expressed on the ex vivo cell preparations and, in parallel, on the histological sections obtained from the same tumor samples and simultaneously stained in the immunofluorescence and immunochemical assays (Figure 1, Figure 3A–C, Appendix A). Therefore, due to the ease of interpretation of the results, the ex vivo analysis yielded the exact number of the marker-positive cancer cells obtained from the whole tumor samples (Table 2). Of note, the biosynthesis of the NSCLC subtype-specific markers was heterogeneous in the cancer cells examined both in each tumor sample and between the different tumor samples tested for patients with one NSCLC subtype (Table 2, Figure 3A,B). In addition, the expression of adenocarcinoma-specific CK7 and squamous cell carcinoma-specific PanCK was acknowledged in some squamous cell carcinoma and adenocarcinoma cells, respectively (Table 2, Figure 3A,B). Interestingly, while AhR and AhRR—the markers that were possibly related to carcinogenesis—were not found to be expressed in the cancer cells of the tumor samples, the expression of CYP1A1 as the main marker of AhR activation [46] was observed in some cancer cells obtained from tumor samples sq1 and ad7 of both smoking and non-smoking patients, respectively (Table 2, Appendix A).

Substantial variations in the results obtained from the ex vivo and histological analyses were observed only in the identification of cancer cells with PD-L1 membrane expression in some tumor samples. A total of 10% and 2% of the PD-L1-positive cancer cells were found on the ex vivo cell preparations obtained from tumor samples sq1 and sq3, respectively, but not on the histological sections obtained from the same tumor samples and simultaneously stained in the immunofluorescence assay using clone ABM4E54 (Abcam) of mouse anti-PD-L1 antibodies (Table 2, Figure 2C). Of note, according to the medical records, PD-L1-positive cancer cells had not been identified in these tumors by a standard immunohistochemical analysis conducted in the Novosibirsk Regional Clinical Oncology Dispensary (Novosibirsk, Russia) using clone SP263 (Ventana) of rabbit anti-PD-L1 antibodies. For the other NSCLC patients, the results of testing PD-L1 were the same in all analyses and had a close identity with data in the medical records.

Thus, ex vivo analysis was more comprehensive and optimal in measuring PD-L1 expression in the cancer cells than histological examination commonly used in routine diagnostic practice. As is known, detection of this biomarker in tumor samples is critically important for choosing immunotherapy in post-operative treatment of NSCLC, especially with lung squamous cell carcinoma.

### 3.4. Immune Cell Landscape in the Tumor Microenvironment and Lung Tissue of NSCLC Patients

The lung adenocarcinoma samples examined by ex vivo analysis after Romanovsky–Giemsa staining had a higher number of immune cells than most lung squamous cell carcinoma samples did (Table 2, Figure 4 and Figure 5), especially immune cells from the myeloid lineage, where macrophages in tumor samples ad1 and ad4-ad8, neutrophils in tumor sample ad2, and eosinophils in tumor sample ad3 were the dominant immune cell populations. A relatively low lymphocyte infiltration was found in most tumor samples examined. As the tumor-associated macrophages and tumor-infiltrating lymphocytes were rare on the histological sections parallelly obtained from the same tumor samples, especially for patients with squamous cell carcinoma (Figure 4), the number of immune cells in the TME was estimated only on the ex vivo cell preparations obtained from the whole tumor samples (Table 2, Figure 5). The ex vivo cell preparations obtained from the attendant lung tissue samples of the same patients were largely composed of alveolar macrophages without any cancer cells and with a minimum number of immune cells related to other types (Table 2).

Notably, all macrophages obtained from the tumor and lung tissue samples of the currently smoking patients had a large number of denser dark inclusions in the cytoplasm and were what is called the smokers’ macrophages (Table 2, Figure 4). The presence of smokers’ macrophages and other types of immune cells in the tumor microenvironment and attendant lung tissue samples of the patients was confirmed by histological examination (Figure 4). Of note, the tobacco smokers’ lung tissue samples were characterized by deep remodeling of lung tissue with the proliferation of fibrous connective tissue in the large areas of the interstitial zones, some loss of alveolar architecture, and large clusters of smokers’ alveolar macrophages within the airspaces of the preserved alveoli, while macrophages without denser dark inclusions in the cytoplasm were observed in tumor and lung tissue samples ad7 and sq3 obtained from the patients who had never smoked and former smokers, respectively (Table 1, Figure 4). At the same time, the smokers’ tumor-associated and alveolar macrophages were observed on all preparations obtained from tumor and lung tissue samples ad6, respectively, of the patient who declared himself as non-smoking (Figure 4).

The tumor-associated neutrophil infiltration was observed mainly in the necrotic areas on the histological sections of tumor samples and thus the increased number of neutrophils on the ex vivo cell preparations indicated the presence of excessive necrotic areas in tumor samples ad2, ad3, and sq1.

Abundant eosinophil infiltration was detected in tumor samples ad3 and sq2, but not in the lung tissue samples of these patients who had no evidence of attendant immune diseases on the ex vivo cell preparations or on the histological sections (Table 2, Figure 4).

Thus, immune cell profiling in the tumor microenvironment revealed significant differences in the immune infiltrates and composition between the tumor samples of the patients with the same NSCLC subtypes in ex vivo and histological analyses performed in parallel.

### 3.5. The Expression Pattern of Immune-Related Markers by Tumor-Associated Macrophages and Alveolar Macrophages

Immune-related marker profiling showed that the tumor-associated macrophages that were CD14-positive expressed both pro-inflammatory, i.e., immunoactivating, IFNγ, TNFα, IL-1β, IL-12, and anti-inflammatory, i.e., immunosuppressive, IL-4 and IL-10 cytokines colocalized in the same cells on the ex vivo cell preparations obtained from all tumor samples examined (Figure 6 and Appendix A). The tumor-associated macrophages also produced the fibroblast growth factor FGFb, the carcinogenesis-related markers AhR, AhRR, CYP1A1, and immune checkpoints, i.e., immunosuppressive molecules, PD-1 localized in intracellular vesicles, and PD-L1 localized both on the plasma membrane and in intracellular vesicles (Figure 6 and Appendix A). For the lung tissue samples, the expression of these markers was observed predominantly in the smokers’ alveolar macrophages obtained from the lung tissues of the smoking patients (Figure 6 and Appendix A). Also, the marker-positive alveolar macrophages were found for the non-smoking patient, whose lung tissue sample ad6 was assayed (Figure 6 and Appendix A). No activated alveolar macrophages were identified in the patients who had never smoked or former smokers, whose lung tissue samples ad7 and sq3, respectively, were examined. However, all alveolar macrophages were positive for CD14 expression. Neither PD-1-positive nor cytokine-producing lymphocytes were detected on the ex vivo cell preparations obtained from the tumor and lung tissue samples of all patients. These results are in an excellent agreement with histological data coming simultaneously from the same tumor and lung tissue samples; however, the tumor-associated macrophages were rare on the histological sections obtained from tumor samples (Appendix A). Therefore, the number of marker-positive macrophages was analyzed only on the ex vivo cell preparations obtained from the whole tumor and lung samples (Figure 6).

Thus, the activated macrophages expressing all markers were detected in the tumor microenvironment for all NSCLC patients and especially in the lung tissue samples of the smoking NSCLC patients.

## 4. Discussion

Lung cancer, including NSCLC, is the leading cause of cancer-related deaths worldwide [1,2] and is etiologically closely associated with tobacco smoking [3]. NSCLC treatment has undergone a major paradigm shift with the advent of targeted therapy using tyrosine kinase inhibitors and checkpoint inhibition immunotherapy [1,2,11], which require biomarkers to be tested to ensure careful patient selection and prediction of clinical benefits from the selected therapy. Now, the molecular and immunohistochemical diagnostics carried out on surgically resected tumor and biopsy specimens are widely used for primary diagnosis and biomarker studies in routine clinical practice. Nevertheless, new diagnostic tools are needed to measure predictive biomarker expression in NSCLC patients with a view to select individual treatment protocols for them. In line with this, on the one hand, the use of cytology specimens [52,53] and circulating tumor cells [54,55] provides information on the expression of predictive biomarkers, including PD-L1, in NSCLC; on the other hand, the spatial metabolomics approach [56] and lipid profiling [57] can contribute to NSCLC subtyping.

In our study, we developed a novel diagnostic method which does not depend on tumor tissue morphology or conventional histology techniques. Our ex vivo analysis is based on (1) ex vivo isolation of cancer and immune cells from the whole samples of surgically resected NSCLC, when these cells are separated from fibrotic stroma and (2) tested on ex vivo cell preparations that are made rapidly, (3) immunostaining and (4) biomarker examination, which are very easily performed on ex vivo cell preparations. Notably, PD-L1 expression was detected only by ex vivo analysis in some NSCLC patients, while the assessment of the other cancer-specific and immune-related markers showed that the results obtained from the ex vivo cell preparations were comparable with those obtained from the histological sections of the same tumor and lung tissue samples for all patients. Thus, in diagnosing PD-L1 that exhibits focal expression in tumor tissues [21,22,23,24,25,26,27,28], ex vivo analysis of NSCLC samples decreased the number of false negatives in the immunohistochemical examination, and so this analysis may be a method of choice in assessing the status of this parameter for cancer cells, especially those with low PD-L1 TPS, and treatment decision making. Currently, the two important PD-L1 TPS therapeutic cutoff points are 1% and 50%, which determine whether a patient can receive a single drug pembrolizumab either as a second-line drug after the progression of platinum-based therapy or as a first-line drug, respectively. In this context, determining the status of PD-L1 in NSCLC patients is very important for their management after surgery.

As is known, the response to anti-NSCLC therapy, including immunotherapy, is a complex process in which factors related to TME heterogeneity have to be taken into account [35,36,37,38]. In this context, the immune cells, such as macrophages, lymphocytes, and others, in the TME might also have an impact on the tumor fate and, therefore, may serve as potential therapeutic targets to reinvigorate anti-tumor immunity [40,42,43,44,45,58,59]. Furthermore, PD-L1 expression on the cancer cell membrane can be induced through the IFNγ and pro-inflammatory immune response [35]. Tumor-associated macrophages are often divided into two sub-populations, showing either only anti-tumoral activity (classical activation with an M1-like phenotype and the expression of pro-inflammatory cytokines that activate adaptive immune response) or, conversely, only pro-tumoral activity (alternative activation with an M2-like phenotype and the expression of anti-inflammatory and immunosuppressive cytokines and growth factors that promote tumor survival and progression) [42,43,44,45,59]. However, now it is becoming clear that activation and polarization of tumor-associated macrophages may consist of a range of non-terminal phenotypes rather than two binary states [43]. With ex vivo analysis, we found that all tumor-associated macrophages were characterized by M1- and M2-like phenotypes simultaneously in all tumor samples examined. In addition, alveolar macrophages with mixed M1/M2 activation were also detected in the lung tissue samples of the smoking patients with both lung adenocarcinoma and squamous cell carcinoma, thereby indicating chronic simultaneous pro- and anti-inflammatory activation. It is possible that the mutagenic microenvironment created by chronic airway inflammation in the smokers’ lungs eventually promotes the development of NSCLCs that originate after the lung epithelial tissue damage. The presence of only smokers’ macrophages in the TME suggests that predominantly alveolar macrophage, rather than blood monocytes and macrophages from other populations found in the human lung [59,60], migrate to tumor tissues of NSCLC. Noteworthy, the expression pattern of the immune-related markers being considered did not vary significantly between different tumor samples both within one NSCLC subtype and across various histological subtypes of NSCLC, while distinct immunologic profiles across tumor types were determined for lung neuroendocrine neoplasms [61]. Overall, our findings are consistent with earlier observations stating that a binary model of tumor-associated macrophage activation may not exist in tumors, including different histological subtypes of NSCLC.

Likewise, ex vivo analysis offered insight into the immune cell landscape of individual NSCLC samples and provided increased efficacy in classifying subpopulations of the tumor infiltrating immune cells in a patient-specific qualitative (identifying cells from the myeloid and lymphoid lineages) and a quantitative (counting immune cells in the entire population) manner. Variable immune cell compositions were identified in the TME of some NSCLC samples, when these tumors (but not the lung tissues) contained—in addition to macrophages, lymphocytes, and neutrophils—a large number of eosinophils revealed by Romanovsky–Giemsa and DAB staining of their intracellular granules, usually appearing as a low-size myeloid cell population. Of note, NSCLC patients with tumor eosinophilia in the TME did not have attendant immune diseases. While a link has been shown between blood eosinophilia and the favorable clinical outcomes of patients after treatment with immune checkpoint inhibitors for advanced lung cancer [62], the exact role and mechanisms of action of the tumor-infiltrating eosinophils in NSCLC has yet to be defined. Nevertheless, eosinophils, similarly to other myeloid cells, may be involved in the immune response against lung cancer. It should be noted that the immune infiltrates of many NSCLC patients were not enriched for lymphocytes and these were characterized by the absence of PD-1 and cytokine expression in all tumor and lung samples tested. Therefore, more clinical studies, including those using our ex vivo analysis, are needed for a detailed understanding of the functions of the cells participating in innate and adaptive immunity in NSCLC patients, and to design accurate strategies for targeting them in the fight against cancer.

Interestingly, while most of the NSCLC patients studied were current smokers, cancer cells with AhR and AhRR expression, which is activated by the many components/carcinogens of tobacco smoke and, therefore, may be used as a biomarker of lung cancer development, were not identified in our work. Consequently, CYP1A1 expression detected in the cancer cells of some NSCLC samples was probably controlled by other transcriptional regulatory pathways that were not linked to activation of AhR signaling. However, further studies focusing on AhR’s involvement in the pathogenesis of NSCLC are required.

Although we have developed a method of ex vivo analysis to assess various characteristics of NSCLC and demonstrated some of its advantages over the routine histological assay, there are still several limitations that need to be addressed in future studies. First, the number of NSCLC samples tested in this work is small, and a larger number of tumor samples is needed to promote ex vivo analysis and validate the results. Secondly, our ex vivo analysis and conclusions concerning the expression of various markers, including PD-L1, and the immunologic landscape, are based on testing only one small part obtained from each surgically resected lung tumor, which is supposed to be highly heterogeneous within different NSCLC tumor areas. In this regard, a larger set of tumor samples from each NSCLC resection should be characterized by ex vivo analysis to contribute to a better understanding of cancer pathogenesis, including the immunological status, and the mechanisms leading to the modulation of PD-L1 expression. Thirdly, in ex vivo analysis, the PD-L1 status of cancer cells is assessed using primary antibodies for biological research, while the other PD-L1 diagnostic immunohistochemical assay systems tailored for each immune checkpoint inhibitor are recommended to determine patient eligibility for immunotherapy treatment of NSCLC in the current clinical setting. Therefore, PD-L1 should be assessed on ex vivo cell preparations with the use of the clinical validated anti-PD-L1 antibody clones for an adequate comparison of distinct assays. Fourthly, because in daily practice more biopsy samples than resection specimens from NSCLC, including advanced-stage cases, are used for diagnosis and biomarker assessment for optimal patient selection to personalize treatment strategies and track therapeutic efficacy, in future studies, tissue sampling should be carried out on biopsies to measure PD-L1 expression in ex vivo analysis. Finally, to evaluate the applicability and performance of ex vivo analysis in NSCLC studies, further studies are required.

## 5. Conclusions

The reported ex vivo analysis allows one to rapidly explore various characteristics of cancer and tumor-infiltrating immune cells without fibrotic stroma after their isolation from surgically resected NSCLC samples and to have a better understanding of the immunological landscape and expression of different cell markers in the tumor as a whole, thereby avoiding some false negatives in the routine histological assay of the same tumor samples. The proposed technical approach may be used as an additional diagnostic tool for measuring PD-L1 scores in cancer cells and making treatment decisions. In summary, reliable diagnostic methods that can accurately assess tumor parameters and identify biomarkers contribute to improved prognostic evaluation and management decisions for NSCLC patients, which is crucial to reducing cancer mortality rates.

## Figures and Tables

**Figure 1 cancers-16-02886-f001:**
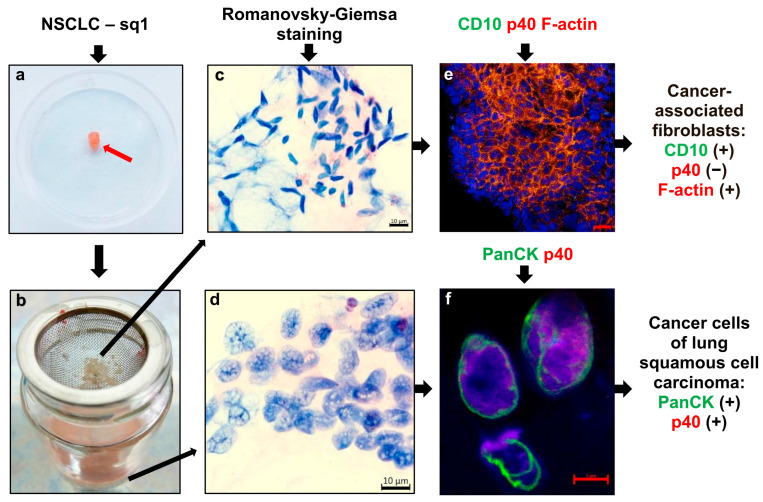
Representative images demonstrate isolation of cancer cells (**a**) from surgically resected tumor sample sq1 (weight 0.07 g), which is indicated by the red arrow in the Petri dish 5 cm in diameter, (**b**) in the cell suspension after separating fibrotic tissue in a sieve and (**c**–**f**) their analysis on (**c**,**e**) cell smears and (**d**,**f**) the ex vivo cell preparations after (**c**,**d**) Romanovsky–Giemsa staining and (**e**,**f**) the immunofluorescence assay with specific antibodies to different NSCLC and fibroblast markers (green and red signals). Nuclei are stained by DAPI (blue signal). (**e**,**f**) Colocalization of the markers is (**e**) yellow and (**f**) magenta (in the nuclei) signals on confocal immunofluorescent images. The scale bars are (**c**,**d**) 10, (**e**) 20, and (**f**) 5 μm.

**Figure 2 cancers-16-02886-f002:**
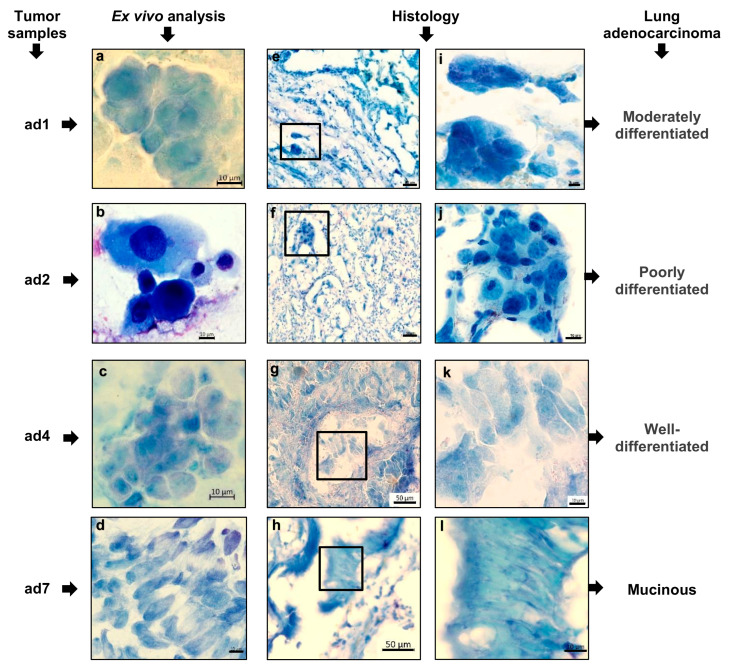
Representative images after Romanovsky–Giemsa staining demonstrate that the differentiation and specific features of the patients’ adenocarcinoma cells and their clusters can be defined not only (**e**–**l**) on the histological sections, but also (**a**–**d**) on the ex vivo cell preparations obtained from the same tumor samples. (**e**–**h**) Close-ups of the parts of the images (**i**–**l**). The scale bars are (**i**) 5, (**a**–**d**,**j**–**l**) 10, and (**e**–**h**) 50 μm.

**Figure 3 cancers-16-02886-f003:**
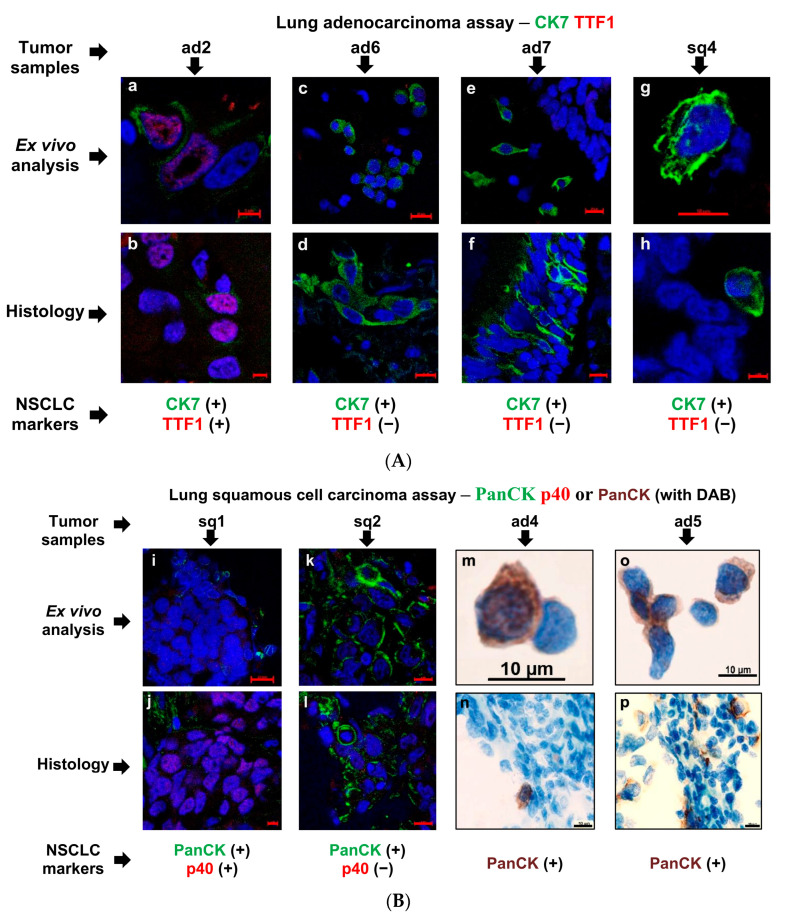
Representative confocal merged immunofluorescent or immunochemical images demonstrate the expression of lung (**A**) adenocarcinoma- and (**B**) squamous-cell-carcinoma-specific markers and (**C**) proliferation marker Ki-67 in cancer cells both on the ex vivo cell preparations and, in parallel, on the histological sections obtained from the same tumor samples, while (**C**) PD-L1 expression is detected (**u**,**w**) in some lung squamous cell carcinoma cells only by ex vivo analysis. Cells and their nuclei are stained with appropriate specific antibodies (green and red signals or brown staining) and DAPI (blue signal), respectively. Localization of some markers in the nuclei is magenta signal. (**t**) Red arrow indicates the anaphase of mitosis. Green arrows indicate the PD-L1-positive cancer cells, as solitary and in clusters. The scale bars are (**a**,**b**,**h**,**j**,**q**,**t**) 5, (**d**,**f**,**g**,**k**–**p**,**r**,**w**) 10, and (**c**,**e**,**i**,**s**,**u**,**v**,**x**) 20 μm.

**Figure 4 cancers-16-02886-f004:**
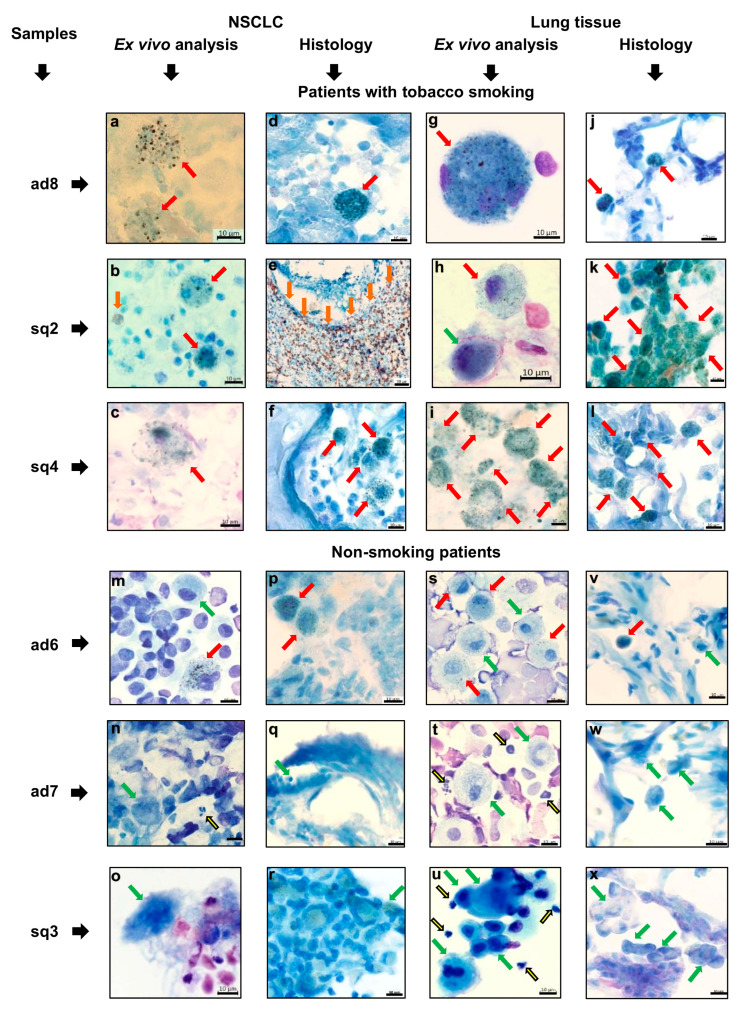
Representative images after Romanovsky–Giemsa staining demonstrate the different types of immune cells detected on the ex vivo cell preparations and, in parallel, on histological sections obtained from the same tumor samples (**a**–**l**) for tobacco smokers and (**m**–**x**) non-smoking patients. Red and green arrows indicate macrophages, as solitary and in clusters, with denser dark inclusions in the cytoplasm (smokers’ macrophages) and without them, respectively. Yellow and brown arrows indicate (**n**,**t**,**u**) neutrophils and (**b**,**e**) eosinophils, respectively, as solitary and in clusters. (**e**) The granules of eosinophils are visualized with DAB substrate. The scale bars are (**a**–**d**,**f**–**x**) 10 and (**e**) 50 μm.

**Figure 5 cancers-16-02886-f005:**
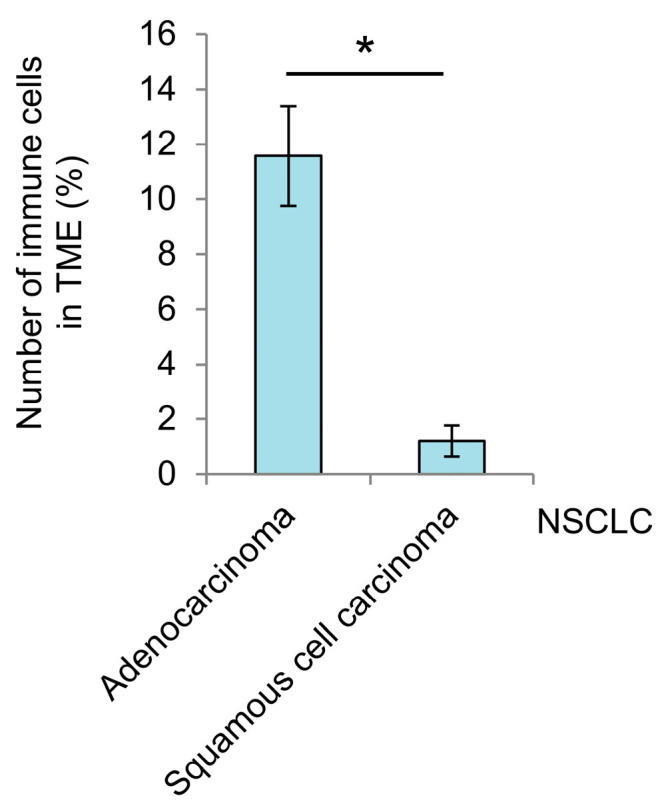
Differences in the number of immune cells in the TME between different NSCLC subtypes are found for the patients without tumor eosinophilia. The total number of immune cells (all types) is expressed as the percentage of the total number of the patients’ cells (cancer and immune) examined on the ex vivo cell preparations for adenocarcinoma ad1, ad3-ad8 (*n* = 7) and squamous cell carcinoma sq1, sq3, sq4 (*n* = 3) samples. Data are expressed as the means ± SEM. * *p* < 0.001.

**Figure 6 cancers-16-02886-f006:**
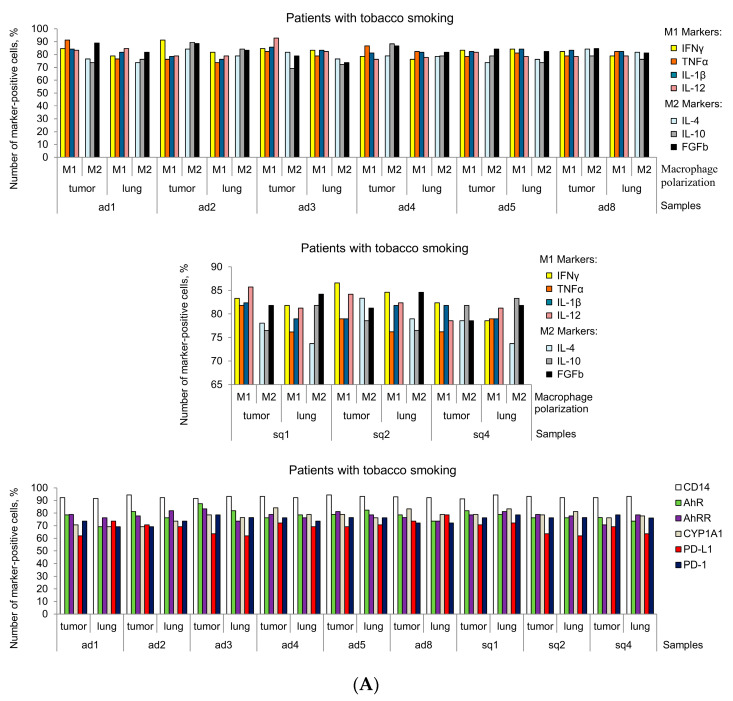
Most macrophages express all markers studied (**A**,**B**) in the tumor microenvironment of all patients’ samples and (**A**) in the lung tissue samples of the smoking NSCLC patients, whereas (**B**) alveolar macrophages only with CD14 expression are identified in the lung tissues for some non-smoking patients. The number of the marker-positive macrophages expressed as the percentage of the total number of the macrophages analyzed on the ex vivo cell preparations.

**Table 1 cancers-16-02886-t001:** Baseline characteristics of NSCLC patients and their tumor samples used in ex vivo analysis.

NSCLC Patients ^1^							
No.	Tumor Samples (g) ^2^	TNM Classification	Sex ^3^	Age (Years)	Smoking Status (Years) ^4^	Attendant Pulmonology Diseases	Surgery ^5^
Adenocarcinoma							
ad1	1.06	T1cN2A2M0(IIIA)	M	72	45		LLL
ad2	0.12	T2aN0M0(IB)	F	68	50	COPD	RLL
ad3	0.19	T1cN0M0(IA3)	M	62	50	Chronic bronchitis	LUL
ad4	0.20	T1cN1M0(IIB)	M	58	45		RLL
ad5	0.17	T2N0M0(IB)	M	63	40		LUL
ad6	0.04	T2aN0M0(IB)	F	55	-		RUL
ad7	0.09	T2bN0M0(IIA)	F	64	-		RLL
ad8	0.12	T1bN0M0(IA2)	M	63	54		RLL
Squamous cell carcinoma							
sq1	0.07	T1cN0M0(IA3)	M	63	45	Chronic bronchitis	RL
sq2	0.13	T1cN0M0(IA3)	M	66	50		LUL
sq3	0.12	T3NxM0	M	67	40 (+), 10 (-)		RLL
sq4	0.12	T1cN1M0(IIB)	M	72	51	COPD, pneumosclerosis	RUL

^1^ Data from medical records. ^2^ Weight of tumor samples for ex vivo analysis. ^3^ M, male; F, female. ^4^ (+), yes; (-), not. ^5^ LLL, left lower lobe; LUL, left upper lobe; RL, right lung; RLL, right lower lobe; RUL, right upper lobe.

**Table 2 cancers-16-02886-t002:** Expression of lung cancer markers and immune cell composition in surgically resected NSCLC and lung tissue samples analyzed on the ex vivo cell preparations.

NSCLC Markers ^1^	NSCLC Patients											
	Adenocarcinoma								Squamous Cell Carcinoma			
	ad1	ad2	ad3	ad4	ad5	ad6	ad7	ad8	sq1	sq2	sq3	sq4
	Lung adenocarcinoma-specific markers											
CK7	5	1	1	1	10	5	2	-	-	nd	nd	0.5
TTF1	-	30	10	-	-	-	-	-	-	nd	nd	-
	Lung squamous-cell-carcinoma-specific markers											
PanCK	5	-	-	5	10	nd	nd	0.5	0.5	1	-	-
p40	nd	nd	nd	nd	nd	nd	nd	-	30	-	-	-
	Proliferation marker											
Ki-67	-	-	-	-	-	-	-	10	50	-	-	20
	Immunotherapy marker											
PD-L1	-	-	-	-	nd	-	-	-	10	-	2	-
PD-L1 (histology) ^2^	-	-	-	-	-	-	-	-	-	-	-	-
	Lung carcinogenesis markers											
AhR	-	-	-	-	-	-	-	nd	-	-	-	nd
AhRR	-	-	-	-	-	-	-	nd	nd	nd	nd	nd
CYP1A1	-	-	-	-	-	-	10	-	10	-	-	-
Tobacco smoking ^3^	+	+	+	+	+	-	-	+	+	+	-	+
	Immune cells ^4^ in the TME											
Macrophages	72.7	1.5	16.1	75	71.4	86.7	58.3	75	33.3	5.8	42.9	87.1
Neutrophils	22.7	95.5	28.6	-	14.3	3.3	16.7	-	66.7	2.5	42.9	-
Lymphocytes	4.6	3	1.2	-	-	10	25	25	-	0.8	14.3	-
Eosinophils	-	-	54.2	25	28.6	-	-	-	-	90.9	-	12.9
Total number ^5^	7.4	41.2	18.4	11.8	13.8	14.2	12.4	3	0.9	28.6	0.3	2.4
Smokers’ macrophages	+	+	+	+	+	+	-	+	+	+	-	+
	Immune cells ^4^ in the lung tissue											
Macrophages	100	54	65.7	82.1	63.9	94	90.2	96.5	44.6	52.5	64.8	99
Neutrophils	-	39.1	16.2	4.1	22.2	1.2	7	3.5	14.3	29.5	31.5	-
Lymphocytes	-	6.9	18.2	-	12.5	4.8	2.8	-	41.1	11.5	3.7	-
Eosinophils	-	-	-	13.8	1.4	-	-	-	-	6.6	-	1
Smokers’ macrophages	+	+	+	+	+	+	-	+	+	+	-	+
Attendant immune diseases ^3^	-	-	-	-	-	-	-	Asthma	-	-	-	-

^1^ The number of the marker-positive cancer cells expressed as the percentage of the total number of the cancer cells examined (for PD-L1 assay: an analog of PD-L1 TPS in immunohistochemistry). ^2^ Analysis of PD-L1 expression on the histological sections obtained, in parallel, from the same tumor samples. For other examinations of the marker-positive cancer cells, the data obtained on the histological sections are similar to those obtained on the ex vivo cell preparations and are not shown in Table. ^3^ Data from medical records. ^4^ Data are presented as the percentage of the number of immune cells of a particular type out of the total number of immune cells examined. ^5^ The total number of immune cells (all types) expressed as the percentage of the total number of the patients’ cells (cancer and immune) examined. (+), is present; (-), is absent; nd, not done.

## Data Availability

The datasets generated for this study are available on request to the corresponding author.

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
