# Peer review of "Development of Ex Vivo Analysis for Examining Cell Composition, Immunological Landscape, Tumor and Immune Related Markers in Non-Small-Cell Lung Cancer"

_cancers, 2024, doi:10.3390/cancers16162886_

Round 1
Reviewer 1 Report
Comments and Suggestions for Authors
Ufimtseva et al. describe a new method to evaluate NSCLC biopsies ex vivo using single cells suspensions from biopsies. They show similarities as well as differences with their histological counterparts. Overall this manuscript is relevant for the field yet a few important assets are missing:
1) No clear numerical side-by-side comparison of all the different markers found within the single cells suspensions and slides. I suggest to add these in a table or even more preferable in histograms to be able to perform statistics.
2) Confusion regarding typing of macrophages. Authors should use specific markers to distinguish alveolar from non-alveolar macrophages or explain how they define them specifically. Overall details regarding types of macrophages that are described in current literature were lacking in results and discussion and should be added.
3) Stainings of immunological markers were not convincing. Figures need to be improved and in addition again compared side-by-side with the respective histological slides to be able to claim similarities.
These and minor comments can also be found within the PDF attached.

Please find my comments and suggestions within the pdf attached.
Author Response
I would like to thank the Reviewer for careful reading of our manuscript and for his/her comments that led to improving of article’s contents.
Comment 1:
“1) No clear numerical side-by-side comparison of all the different markers found within the single cells suspensions and slides. I suggest to add these in a table or even more preferable in histograms to be able to perform statistics.”
Answer for Comment 1:
Now, in the Results, in lines 466-470: The Figure 6 with the summarized results of the markers expression in histograms are shown.
Comment 2:
“2) Confusion regarding typing of macrophages. Authors should use specific markers to distinguish alveolar from non-alveolar macrophages or explain how they define them specifically. Overall details regarding types of macrophages that are described in current literature were lacking in results and discussion and should be added.”
Answer for Comment 2:
Now, in the Results, in lines 421-429: “…Of note, the tobacco smokers' lung tissue samples were characterized by deep remodeling of lung tissue with the proliferation of fibrous connective tissue in the large areas of the interstitial zones, some loss of alveolar architecture, and large clusters of smokers’ alveolar macrophages within the airspaces of the preserved alveoli, while macrophages without denser dark inclusions in the cytoplasm were observed in tumor and lung tissue samples ad7 and sq3 obtained from the never and former smokers, respectively (Table 1, Figure 4). At the same time, the smokers’ tumor-associated and alveolar macrophages were observed on all the preparations obtained from tumor and lung tissue samples ad6, respectively, of the patient who declared himself as non-smoking (Figure 4).”
Now, in the Discussion, in lines 530-533: “The presence of only smokers’ macrophages in the TME suggests that predominantly alveolar macrophage, rather than blood monocytes and macrophages from other populations found in the human lung [59,60], migrate to tumor tissues of NSCLC.”
In our study, the macrophages obtained from tumor samples in ex vivo analysis were classified as “tumor-associated macrophages”. The smokers’ macrophages with a large number of denser dark inclusions in the cytoplasm were obtained from lung tissue samples of patients in ex vivo analysis and reside in alveoli on the histological sections. Therefore, they were classified as “alveolar macrophages”.
In the Results, in lines 416-421: “Notably, all macrophages obtained from the tumor and lung tissue samples of the currently smoking patients had a large number of denser dark inclusions in the cytoplasm and were what is called the smokers’ macrophages (Table 2, Figure 4). The presence of smokers’ macrophages and other types of immune cells in the tumor microenvironment and attendant lung tissue samples of the patients was confirmed by histological examination (Figure 4).”
Comment 3:
“3) Stainings of immunological markers were not convincing. Figures need to be improved and in addition again compared side-by-side with the respective histological slides to be able to claim similarities.”
Answer for Comment 3:
Now, in the Results, in lines 466-470: Figure 6. The most macrophages express all the markers studied (A, B) in the tumor microenvironment of all the patients’ samples and (A) in the lung tissue samples of the smoking NSCLC patients, whereas (B) alveolar macrophages only with CD14 expression are identified in the lung tissues for some non-smoking patients. The number of the markers-positive macrophages expressed as the percentage of the total number of the macrophages analyzed on the ex vivo cell preparations.
Now, in the Supplementary Materials, in lines 30-42: Figure S3. Representative confocal immunofluorescent images demonstrate the presence of immune markers expression in the most tumor-associated macrophages and alveolar macrophages with denser dark inclusions in the cytoplasm obtained from the tumor and lung tissues, respectively, for (A – a-x) smoking and (B – a-h, q, s, u, w) non-smoking (according to medical records) patients, but (B – i, l, n, p, r, t, v, x) its absence in the alveolar macrophages without denser dark inclusions in the cytoplasm obtained from the lung tissues of some non-smoking patients on the ex vivo cell preparations and, in parallel, histological sections…
Comment 4:
“4) These and minor comments can also be found within the PDF attached.”
Answer for Comment 4:
- Now, in the Abstract, in line 23: “…separation…”
- Now, in the Abstract, in line 23: “…separation…”
- Now, in the Abstract, in line 30: “…assays have…”
- Now, in the Abstract, in lines 40-41: “…examiniation of…”
- Now, in the Introduction, in line 53: “…improve…”
- Now, in the Introduction, in line 127: “…assays…”
- Now, in the Introduction, in line 128: “…have…”
- Now, in the Introduction, in line 131: “…for…”
- Now, in the Materials and Methods, in lines 173-174: “…without any enzymatic treatments.”
- Answer for Comment “From which company was the sieve purchased?”: In our work, we used the ordinary tea strainers (made in China) from a supermarket.
- Answer for Comment “It is not clear from Table 2 that the exact number of marker-positive cancer cells was obtained as in the histological samples. I suggest to add within the table all the numbers that were obtained for the same markers from the histological sections (as the authors did for PD-L1 expression). This is the only way to be able to compare all different markers side by side.”: We decided not to overload the Table 2 with the data, because (in the Results, in lines 330-335) “Detection of tumor-specific markers revealed similar numbers of cancer cells that were positive for…markers expressed on the ex vivo cell preparations and, in parallel, on the histological sections obtained from the same tumor samples…” Now, in the Results, in lines 336-338: “Therefore, due to the ease of interpretation of the results, the ex vivo analysis yielded the exact number of the marker-positive cancer cells obtained from the whole tumor samples (Table 2).” Now, in Table 2, in lines 362-366: “Analysis of PD-L1 expression on the histological sections obtained, in parallel, from the same tumor samples. For other examinations of the marker-positive cancer cells, the data obtained on the histological sections are similar to those obtained on the ex vivo cell preparations and are not shown in the Table.”
- Now, in the Results, in line 345: “…expressed…”
- Answer for Comment “Also for this figure, the data of the histological samples should be included to be able to compare both data.”: Now, in the Results, in lines 394-398: “As the tumor-associated macrophages and tumor-infiltrating lymphocytes were rare on the histological sections parallelly obtained from the same tumor samples, especially for patients with squamous cell carcinoma (Figure 4), the number of immune cells in the TME was estimated only on the ex vivo cell preparations obtained from the whole tumor samples (Table 2, Figure 5).”
- Now, in the Results, in line 418: “…Figure 4…”
- Now, in the Results, in line 421: “…Figure 4…”
- Answer for Comment “It is unclear how the authors differentiate tumor associated macrophages from alveolar macrophages. Especially as they state that they both express CD14. In lung cancer different types of macrophages are present (both monocyte-derived and alveolar macrophages) while the authors seem to state that in lung cancer you only have TAMs and in healthy tissue you only have AMs, while lung cancer also holds AMs. Can therefore a AM specific marker be used to distinguish the TAMs from the AMs? e.g. PPARg or MARCO for AMs (cfr. https://pubmed.ncbi.nlm.nih.gov/34659209/ and https://pubmed.ncbi.nlm.nih.gov/34767762/)”: In our study, the macrophages obtained from tumor samples in ex vivo analysis were classified as “tumor-associated macrophages”. The smokers’ macrophages with a large number of denser dark inclusions in the cytoplasm were obtained from lung tissue samples of patients in ex vivo Also, the smokers’ macrophages are located in alveoli on the histological sections of the same lung tissue samples. Therefore, they were classified as “alveolar macrophages”. CD14 marker is a cell surface marker for the identification and characterization of human macrophage/monocyte (Figure from BD_CD Marker Handbook).
Of course, it is necessary to find the markers of TAMs that will distinguish them from alveolar macrophages from lung tissue. Now, in the Results, in lines 443-444: “Immune-related marker profiling showed that the tumor-associated macrophages that were CD14-positive expressed both pro-inflammatory…”
- Answer for Comment “Figure S5 is not present. Don't the authors mean S3? Secondly, the stainings of the immune markers in green are not clear at all from the pictures. Authors should show clearer images or redo the analysis. Also, ex vivo analysis allows the analysis of the cells with flow cytometry, giving a much more quantitative idea of the % of cells that are positive for a specific marker. Can this be performed as well?”: Now, in the Results, in line 447: “…Figures 6 and S3…”, in lines 460-464: “…These results are in an excellent agreement with histological data coming simultaneously from the same tumor and lung tissue samples, however, the tumor-associated macrophages were rare on the histological sections obtained from tumor samples (Figure S3). Therefore, the number of the markers-positive macrophages was analyzed only on the ex vivo cell preparations obtained from the whole tumor and lung samples (Figure 6).” In our work, we did not use flow cytometry, because the quantity of the tumor-associated macrophages obtained for ex vivo analysis was insufficienttousethis
- Answer for Comment “ex vivo 'expansion' is confusing. As far as I understand, the tissue was only passed through a sieve without subsequent cultivation (and expansion). Hence isolation/purification seems more appropriate here”: Now, in the Discussion, in line 491: “…ex vivo isolation…” Also, in the Abstract, in line 32: “…ex vivo isolation…”, in the Results, in line 287: “…ex vivo isolation…”, in the Conclusions, in line 595: “…isolation…”
- Answer for Comment “To be able to conclude this, a clear side-by-side comparisison needs to be performed, preferrably depicted in histograms instead of a table to show significances as well.”: Now, in the Supplementary Materials, in lines 30-42: Figure S3. Representative confocal immunofluorescent images demonstrate the presence of immune markers expression in the most tumor-associated macrophages and alveolar macrophages with denser dark inclusions in the cytoplasm obtained from the tumor and lung tissues, respectively, for (A – a-x) smoking and (B – a-h, q, s, u, w) non-smoking (according to medical records) patients, but (B – i, l, n, p, r, t, v, x) its absence in the alveolar macrophages without denser dark inclusions in the cytoplasm obtained from the lung tissues of some non-smoking patients on the ex vivo cell preparations and, in parallel, histological sections…
- Answer for Comment “Again, authors can not state this as no alveolar versus monocyte-derived macrophage markers were used.”: Now, in the Discussion, in lines 530-533: “The presence of only smokers’ macrophages in the TME suggests that predominantly alveolar macrophage, rather than blood monocytes and macrophages from other populations found in the human lung [59,60], migrate to tumor tissues of NSCLC.”
- Now, in the Discussion, in line 553: “…defined.”
I’m thankful to the Reviewer for English Language and feel sorry for the mistakes made during the Russian-English translation.
Once again, I would like to thank the Reviewer for careful reading of our manuscript and for their comments that led to improving of article’s contents.
Elena Ufimtseva,
PhD, Federal Research Center of Fundamental and Translational Medicine, Novosibirsk, Siberia, Russia.

Reviewer 2 Report
Comments and Suggestions for Authors
The authors that examine the NSCLC samples from patients that traced the sample for testing cancer and immune cells in whole tumor samples. They try to establish ex vivo approach for amplifying the cancers for further study. Furthermore, by using this system quick examining cancer and immune cells in whole tumor samples and avoiding false negatives in histological assays. It is interesting, some comments as following:
1. In the title have mention the cytokine tracing such as IL-4, IL-10 and IL-12 et al., but in the results did not showed in the Figures.
2. I prefer to see a summary figure for summarized the process and results.
3. The references should be updated on 2024.
4. The scale bars in the figures should be labeled and more clear.
Comments on the Quality of English LanguageMinor editing of English language required.
Author Response
I would like to thank the Reviewer for careful reading of our manuscript and for his/her comments that led to improving of article’s contents.
Comment 1:
- “In the title have mention the cytokine tracing such as IL-4, IL-10 and IL-12 et al., but in the results did not showed in the Figures.”
Answer for Comment 1:
Now, in the Supplementary Materials, in Figure S3: The IL-4-, IL-10-, и IL-12-positive macrophages are shown.
Comment 2:
- “I prefer to see a summary figure for summarized the process and results.”
Answer for Comment 2:
Now, in the Results, in lines 466-470: The Figure 6 with the summarized results of the markers expression in histograms are shown.
Comment 3:
- “The references should be updated on 2024.”
Answer for Comment 3:
Now, in the References,
in lines 668-670:
“12. Illini, O.; Saalfeld, F.C.; Christopoulos, P.; Duruisseaux, M.; Vikstrom, A.; Peled, N.; Demedts, I.; Dudnik, E.; Eisert, A.; Hashemi, S.M.S.; et al. Mobocertinib in patients with EGFR exon 20 insertion-positive non-small cell lung cancer (MOON): An international real-world safety and efficacy analysis. Int. J. Mol. Sci. 2024, 25, 3992. DOI: 10.3390/ijms25073992”.
In lines 740-742:
“39. Cords, L.; Engler, S.; Haberecker, M.; Rüschoff, J.H.; Moch, H.; de Souza, N.; Bodenmiller, B. Cancer-associated fibroblast phenotypes are associated with patient outcome in non-small cell lung cancer. Cancer Cell 2024, 42, 396–412. DOI: 10.1016/j.ccell.2023.12.021”.
In lines 792-793:
“59. Zhou, Y.; Qian, M.; Li, J.; Ruan, L.; Wang, Y.; Cai, C.; Gu, S.; Zhao, X. The role of tumor-associated macrophages in lung cancer: From mechanism to small molecule therapy. Biomed. Pharmacother. 2024, 170, 116014. DOI: 10.1016/j.biopha.2023.116014”.
Comment 4:
- “The scale bars in the figures should be labeled and more clear.”
Answer for Comment 4:
Now, in the Results and Supplementary Materials, in Figures 1, 2, 3A-C, 4 and S1, S2, S3A-B: The scale bars are clearly visible on all cell images.
Comment 5:
“Comments on the Quality of English Language
Minor editing of English language required.”
Answer for Comment 5:
I’m thankful to the Reviewer for this comment and feel sorry for the mistakes made during the Russian-English translation. Now, I corrected some language mistakes, but I rely on the help of the editors of Journal to correct other unseen language mistakes.
Once again, I would like to thank the Reviewer for careful reading of our manuscript and for their comments that led to improving of article’s contents.
Elena Ufimtseva,
PhD,
Federal Research Center of Fundamental and Translational Medicine,
Novosibirsk, Siberia, Russian Federation.

Reviewer 3 Report
Comments and Suggestions for Authors
INTRODUCTION
- The phrase "the major risk factor for this disease" in line 49 is redundant because "tobacco smoking" is already mentioned as the major risk factor.
- Repetitive mentioning of immune checkpoint inhibitors and PD-L1's role (lines 73-91).
- The sentence "In contrast, rarely do lung squamous cell carcinomas harbor these driver mutations and/or gene translocations" (lines 68-69) could be clearer by rephrasing to "Lung squamous cell carcinomas rarely harbor these driver mutations or gene translocations."
- The sentence starting with "Today, the advent of immunotherapy with immune checkpoint inhibitors targeting the PD-1 receptor and its PD-L1 ligand..." (lines 73-74) can be simplified for better readability.
- The phrase "novel mutations of these and other genes" (line 70) should be corrected to "novel mutations in these and other genes."
- Abbreviations should be defined at first use and used consistently. For example, "TTF1"
- The references in the text are not consistently formatted, sometimes appearing as "[1,2]" and other times as "[7-10]." Ensure a consistent format is used throughout.
- The term "poorly differentiated NSCLC" (line 59) should be clarified for readers who may not be familiar with the terminology.
- The phrase "long test duration" (line 128) is vague. It would be better to specify the approximate duration.
- The sentence "Consequently, tumor samples with very large fibrotic areas and the focal nature of marker expression and immune cell composition may yield biased information regarding markers, cell compositions and immunological landscapes." (lines 130-132) is missing a subject after "Consequently."
- The explanation of the PD-L1 tumor proportion score (TPS) (lines 82-85) needs to be clearer. It would be helpful to specify how TPS is calculated and its clinical significance.
- The detailed description of the AhR pathway (lines 109-124) could be made more concise and directly related to its impact on NSCLC.
- The paragraph discussing PD-L1 and immunotherapy responses (lines 73-93) could be better structured to improve logical flow. Consider grouping related points together to enhance coherence.
- The transition between discussing general NSCLC characteristics and specific details about PD-L1 and immunotherapy could be smoother.
- The background on the role of AhR and its regulatory mechanisms (lines 109-122) may be too detailed for an introduction and could be summarized or moved to a later section.
- "Immunochemical staining" should be "immunohistochemical staining" (line 127).
- "Fibrotic tissue in a tumor stroma" should be "fibrotic tissue in the tumor stroma" (line 139).
METHODOLOGY
- The approval reference number “15/2021/11/16” should be checked for format and accuracy according to the institution's standard. It may need to be confirmed or formatted correctly.
- Section 2.1: There is redundancy in specifying “patients’ cells” twice when discussing the division of samples. This could be more concise.
- Section 2.2: The phrase “rubbing through a metal screen of a sieve with pores 0.5-1.0 mm in diameter” lacks clarity. The process and rationale behind choosing this method need better explanation. Alternative, more standardized methods for cell dissociation should be considered or justified.
- Cell Pellet Handling: The duration for centrifugation (5 minutes) and the temperature (room temperature) are mentioned but not the rationale. The centrifuge's speed and time should be validated to ensure cell integrity and viability.
- Section 2.3: The description of antibodies and staining processes is detailed, but it would be beneficial to mention the dilutions used for the primary and secondary antibodies explicitly. Consistency in specifying the sources of antibodies is essential; for instance, Abcam (England) should be formatted uniformly.
- Section 2.3: Fixation times and conditions for different antibodies and stains need to be optimized and justified. The use of 4% formaldehyde and 0.3% Triton X-100 should be cross-validated with literature to ensure consistency with established protocols.:
- Section 2.4: The fixation process duration (15 hours at +4 °C) might be too long, potentially affecting antigenicity. This should be compared with other standard methods.
- The cryosectioning at -25 °C should specify the rationale for choosing this temperature and its consistency with tissue preservation standards.
- Section 2.5: The description of the microscopy settings (objectives, software) is adequate, but the method of quantifying and analyzing the data should be detailed further. The criteria for “analyzing more than 800 cancer cells” should be defined clearly.
- Section 2.6: The mention of Prism 6.0 and Microsoft Excel 2010 is good, but it should also include how the data were normalized or if any specific statistical tests were used in addition to Student’s t-test. Clarify how the P-values were adjusted for multiple comparisons if applicable.
- The formatting of section titles and subsections should be consistent (e.g., capitalization, bold/italic formatting).
- The methodology could benefit from subsections or bullet points to enhance readability, especially in detailed protocols like staining and sample preparation.
- Ensure that all references, especially for reagents and methods, are up-to-date and accurately cited. Some references, like “Romanovsky–Giemsa stain,” need proper citation for the protocol used.
- Section 2.3: The use of Hep-2 cells as internal controls should be explained in more detail, specifically why they were chosen and how they validate the staining process.
RESULTS
· In Table 1, the unit for tumor sample weight (g) is not consistently labeled for each entry, leading to potential confusion about the exact measurements.
· There is a lack of detailed statistical analysis or justification for the sample size (n=12) used in the study, which may affect the robustness and generalizability of the conclusions.
· The descriptions of Figures 1 and 2 lack clarity and precision. For example, it is not clear what specific cellular features are being highlighted or how the images support the conclusions drawn.
· The paper mentions specific markers such as TTF1, CK7, p40, and PanCK, but does not provide detailed methodology on how these markers were quantified or the statistical significance of their expression levels.
· The study does not clearly state the use of control samples, making it difficult to determine the baseline against which the tumor samples were compared.
· In Table 2, the presentation of data on immune cell composition is confusing, with percentages and raw numbers intermixed without clear explanation. This makes it difficult to interpret the data accurately.
· The discrepancy between ex vivo and histological analysis of PD-L1 expression is noted, but the reason for this discrepancy is not explored in detail, which is crucial for understanding the validity of the findings.
· The paper claims that ex vivo analysis is more comprehensive than histological examination for measuring PD-L1 expression, but it does not provide sufficient evidence or rationale to support this claim.
· There is no mention of data normalization or adjustment for potential confounding factors such as age, smoking status, or concomitant diseases, which could bias the results.
· The methodology section lacks details on the protocols used for cell preparation, staining, and analysis, which are crucial for reproducibility and validation of the findings.
Comments on the Quality of English Language
Author Response
I would like to thank the Reviewer for careful reading of our manuscript and for his/her comments that led to improving of article’s contents.
Comment 1:
- “The phrase "the major risk factor for this disease" in line 49 is redundant because "tobacco smoking" is already mentioned as the major risk factor.”
Answer for Comment 1:
The passage in the Introduction, in lines 49-50 “…with tobacco smoking being the major risk factor for this disease…” is the first about tobacco smoking in NSCLC development, therefore, we did not change this phrase. However, now, in the Discussion, in lines 556-557: “Interestingly, while most of the NSCLC patients studied were current smokers, cancer cells with AhR and AhRR expression…”
Comment 2:
- “Repetitive mentioning of immune checkpoint inhibitors and PD-L1's role (lines 73-91).”
Answer for Comment 2:
The analysis of PD-L1 molecules is very important for the treatment of patients with NSCLC, so we have devoted a large paragraph to PD-L1 and its role in immunology and immunotherapy.
Comment 3:
- “The sentence "In contrast, rarely do lung squamous cell carcinomas harbor these driver mutations and/or gene translocations" (lines 68-69) could be clearer by rephrasing to "Lung squamous cell carcinomas rarely harbor these driver mutations or gene translocations."”
Answer for Comment 3:
Now, in the Introduction, in lines 68-69: “In contrast, lung squamous cell carcinomas rarely harbor these driver mutations and/or gene translocations.”
Comment 4:
- “The sentence starting with "Today, the advent of immunotherapy with immune checkpoint inhibitors targeting the PD-1 receptor and its PD-L1 ligand..." (lines 73-74) can be simplified for better readability.”
Answer for Comment 4:
Now, in the Introduction, in lines 73-74: “Today, the advent of immunotherapy with immune checkpoint inhibitors targeting the PD-1 receptor and its PD-L1 ligand has become a great help for NSCLC patients…”
Comment 5:
- “The phrase "novel mutations of these and other genes" (line 70) should be corrected to "novel mutations in these and other genes."”
Answer for Comment 5:
Now, in the Introduction, in line 70: “novel mutations in these and others genes…”
Comment 6:
- “Abbreviations should be defined at first use and used consistently. For example, "TTF1"”
Answer for Comment 6:
I corrected abbreviations in the text and figures.
Comment 7:
- “The references in the text are not consistently formatted, sometimes appearing as "[1,2]" and other times as "[7-10]." Ensure a consistent format is used throughout.”
Answer for Comment 7:
I have rechecked all the References and formatted them according to the journal's guidelines.
Comment 8:
- “The term "poorly differentiated NSCLC" (line 59) should be clarified for readers who may not be familiar with the terminology.”
Answer for Comment 8:
Now, in the Introduction, in lines 59-60: “…poorly differentiated NSCLC with a high degree of cell pleomorphism …”
Comment 9:
- “The phrase "long test duration" (line 128) is vague. It would be better to specify the approximate duration.”
Answer for Comment 9:
Now, in the Introduction, in lines 128-129: “…a long test duration (4-7 days)…”
Comment 10:
- “The sentence "Consequently, tumor samples with very large fibrotic areas and the focal nature of marker expression and immune cell composition may yield biased information regarding markers, cell compositions and immunological landscapes." (lines 130-132) is missing a subject after "Consequently."”
Answer for Comment 10:
Now, in the Introduction, in line 131: “Also, tumor samples with very large fibrotic areas…”
Comment 11:
- “The explanation of the PD-L1 tumor proportion score (TPS) (lines 82-85) needs to be clearer. It would be helpful to specify how TPS is calculated and its clinical significance.”
Answer for Comment 11:
Now, in the Introduction, in lines 83-86: “…evaluation of the PD-L1 tumor proportion score (TPS), which measures the percentage of marker-expressing cancer cells, are widely used as a predictive biomarker assay for anti-PD-1/PD-L1 therapies for NSCLC patients in routine clinical practice [17,19,20].”
Now, in the Discussion, in lines 503-507: “Currently, the two important PD-L1 TPS therapeutic cutoff points are 1% and 50%, which determine whether a patient can receive a single drug pembrolizumab either as a second-line drug after the progression of platinum-based therapy or as a first-line drug, respectively. In this context, determining the status of PD-L1 in NSCLC patients is very important for their management after surgery.”
Comment 12:
- “The detailed description of the AhR pathway (lines 109-124) could be made more concise and directly related to its impact on NSCLC.”
Answer for Comment 12:
As tobacco smoking is the major risk factor for NSCLC, understanding the role of the AhR/AhRR/CYP1A1 axis in lung cancer progression is of importance to improve current therapeutic approaches and develop new therapeutic strategies for the treatment of this disease. I did not shorten this paragraph.
Comment 13:
- “The paragraph discussing PD-L1 and immunotherapy responses (lines 73-93) could be better structured to improve logical flow. Consider grouping related points together to enhance coherence.”
Answer for Comment 13:
Thank the Reviewer for this comment, but I did not change this paragraph, because we are discussing various aspects and problems of PD-L1 examination in NSCLC in them.
Comment 14:
- “The transition between discussing general NSCLC characteristics and specific details about PD-L1 and immunotherapy could be smoother.”
Answer for Comment 14:
I did not change the text in the Introduction.
Comment 15:
- “The background on the role of AhR and its regulatory mechanisms (lines 109-122) may be too detailed for an introduction and could be summarized or moved to a later section.”
Answer for Comment 15:
As tobacco smoking is the major risk factor for NSCLC, understanding the role of the AhR/AhRR/CYP1A1 axis in lung cancer progression is of importance to improve current therapeutic approaches and develop new therapeutic strategies for the treatment of this disease. I did not shorten this paragraph.
Comment 16:
- “"Immunochemical staining" should be "immunohistochemical staining" (line 127).”
Answer for Comment 16:
Now, in the Introduction, in line 127: “…assays with conventional and immunohistochemical staining of histological preparations…”
Comment 17:
- “"Fibrotic tissue in a tumor stroma" should be "fibrotic tissue in the tumor stroma" (line 139).”
Answer for Comment 17:
Now, in the Introduction, in line 139: “…fibrotic tissue in the tumor stroma…”
Comment 18:
- “The approval reference number “15/2021/11/16” should be checked for format and accuracy according to the institution's standard. It may need to be confirmed or formatted correctly.”
Answer for Comment 18:
In the Materials and Methods, in lines 157-158: “This study was approved by the Ethical Committee of the Novosibirsk Regional Clinical Oncology Dispensary (15/2021/11/16).” It is true. The scan of this document from the Ethical Committee of the Novosibirsk Regional Clinical Oncology Dispensary (in Russian Language) was sent to the editorial office of Cancers.
Comment 19:
- “Section 2.1: There is redundancy in specifying “patients’ cells” twice when discussing the division of samples. This could be more concise.”
Answer for Comment 19:
I did not shorten this passage for a better understanding of our procedures in this work.
Comment 20:
- “Section 2.2: The phrase “rubbing through a metal screen of a sieve with pores 0.5-1.0 mm in diameter” lacks clarity. The process and rationale behind choosing this method need better explanation. Alternative, more standardized methods for cell dissociation should be considered or justified.”
Answer for Comment 20:
Now, in the Materials and Methods, in lines 170-174: “In this work, samples were cut into small pieces and, to separate cell suspension containing cancer and immune cells from fibrotic tissue, were further rubbed through a metal screen of a sieve with pores 0.5-1.0 mm in diameter in phosphate-buffer saline (PBS, pH 7.4) without any enzymatic treatments.”
Comment 21:
- “Cell Pellet Handling: The duration for centrifugation (5 minutes) and the temperature (room temperature) are mentioned but not the rationale. The centrifuge's speed and time should be validated to ensure cell integrity and viability.”
Answer for Comment 21:
In our work with using the cytocentrifuge Awel MF20 (Domel, Slovenia), the duration for centrifugation (5 minutes) and room temperature were optimal for obtaining thin-layer cell preparations with the maintenance of cellular integrity and the transfer of cells into a monolayer (see Figures 1, 2, 3A-C, 4, and S1, S2, S3A-B).
Comment 22:
- “Section 2.3: The description of antibodies and staining processes is detailed, but it would be beneficial to mention the dilutions used for the primary and secondary antibodies explicitly. Consistency in specifying the sources of antibodies is essential; for instance, Abcam (England) should be formatted uniformly.”
Answer for Comment 22:
In the Materials and Methods, in line 205: “…diluted 1:100 each.”, in line 210: “…diluted 1:400 each.” Also, I formatted the sources of antibodies uniformly.
Comment 23:
“Section 2.3: Fixation times and conditions for different antibodies and stains need to be optimized and justified. The use of 4% formaldehyde and 0.3% Triton X-100 should be cross-validated with literature to ensure consistency with established protocols.:”
Answer for Comment 23:
The use of 4% formaldehyde and 0.3% Triton X-100 is a standard for cell immunostaining with antibodies [In references, see 50].
Comment 24:
- “Section 2.4: The fixation process duration (15 hours at +4 °C) might be too long, potentially affecting antigenicity. This should be compared with other standard methods.”
Answer for Comment 24:
In the Materials and Methods, in lines 179-181: “The ex vivo cell preparations were immediately fixed with 4% formaldehyde solution in PBS for 10 minutes at room temperature.”, in lines 229-230: “The other portion {for histology} was fixed with 4% formaldehyde solution in PBS for 15 hours at +4 °C.” – these procedures and their duration are standard for cytology and histology, respectively.
Comment 25:
- “The cryosectioning at -25 °C should specify the rationale for choosing this temperature and its consistency with tissue preservation standards.”
Answer for Comment 25:
This temperature is optimal for obtaining frozen sections of lung tissues (according to the recommendations of the manufacturer for Microtome Cryostat HM550 (Microm, Germany). See also: Ufimtseva et al. Mycobacterium tuberculosis Load in Host Cells and the Antibacterial Activity of Alveolar Macrophages are Linked and Differentially Regulated in Various Lung Lesions of Patients with Pulmonary Tuberculosis. Int. J. Mol. Sci. 2021, 22, 3452. https://doi.org/10.3390/ijms22073452 and Ufimtseva and Eremeeva. Drug-Tolerant Mycobacterium tuberculosis Adopt Different Survival Strategies in Alveolar Macrophages of Patients with Pulmonary Tuberculosis. Int. J. Mol. Sci. 2023, 24, 14942. https://doi.org/10.3390/ijms241914942
Comment 26:
- “Section 2.5: The description of the microscopy settings (objectives, software) is adequate, but the method of quantifying and analyzing the data should be detailed further. The criteria for “analyzing more than 800 cancer cells” should be defined clearly.”
Answer for Comment 26:
Now, in the Materials and Methods, in lines 257-259: “All cancer cells (from 800 to 2000 cells) were analyzed in each ex vivo cell preparation.”
Comment 27:
- “Section 2.6: The mention of Prism 6.0 and Microsoft Excel 2010 is good, but it should also include how the data were normalized or if any specific statistical tests were used in addition to Student’s t-test. Clarify how the P-values were adjusted for multiple comparisons if applicable.”
Answer for Comment 27:
Thank the Reviewer for comment, but we did not use any specific statistical tests in this study.
Comment 28:
- “The formatting of section titles and subsections should be consistent (e.g., capitalization, bold/italic formatting).”
Answer for Comment 28:
I have formatted section titles and subsections according to the journal's guidelines.
Comment 29:
- “The methodology could benefit from subsections or bullet points to enhance readability, especially in detailed protocols like staining and sample preparation.”
Answer for Comment 29:
Now, in the Materials and Methods, in lines 184-224 (section 2.3) and in lines 226-246 (section 2.4): the text was divided into several paragraphs.
Comment 30:
- “Ensure that all references, especially for reagents and methods, are up-to-date and accurately cited. Some references, like “Romanovsky–Giemsa stain,” need proper citation for the protocol used.”
Answer for Comment 30:
In the Materials and Methods, in lines 184-187: “To visualize cancer cells and different types of immune cells, some ex vivo cell preparations were washed with PBS and stained with a mixture of azure, eosin, and methylene blue in Romanovsky–Giemsa stain (Minimed, Moscow, Russia) according to the manufacturer’s instruction.”
In Wikipedia:
“Romanowsky stain
From Wikipedia, the free encyclopedia
Romanowsky staining is a prototypical staining technique that was the forerunner of several distinct but similar stains widely used in hematology (the study of blood) and cytopathology (the study of diseased cells). Romanowsky-type stains are used to differentiate cells for microscopic examination in…
Types
Giemsa stain
Giemsa stain is composed of "Azure II" and eosin Y with methanol and glycerol as the solvent.[15] "Azure II" is thought to be a mixture of azure B (which Giemsa called "azure I") and methylene blue…
Giemsa G. Eine Vereinfachung und Vervollkommnung meiner Methylenazur-Methylenblau-Eosin-Färbemethode zur Erzielung der Romanowsky-Nochtschen Chromatinfärbung. // Centralbl f Bakt etc : magazin. — 1904. — Bd. 37. — S. 308—311.” (in German language)
In Russian Federation, Romanovsky–Giemsa stain is standard for clinical practice and is produced by many biotech companies. (In Russian: Романовский – to English: Romanovsky)
Comment 31:
- “Section 2.3: The use of Hep-2 cells as internal controls should be explained in more detail, specifically why they were chosen and how they validate the staining process.”
Answer for Comment 31:
Now, in the Materials and Methods, in lines 213-216: “Actively growing human larynx carcinoma Hep-2 cells (courtesy of PhD. M.V. Solomatina of the Federal Research Center of Fundamental and Translational Medicine, Novosibirsk, Russia) were used as the internal control to validate the adequacy of the Ki-67 staining reaction…”
Comment 32:
“In Table 1, the unit for tumor sample weight (g) is not consistently labeled for each entry, leading to potential confusion about the exact measurements.”
Answer for Comment 32:
In the Results, in Table 1, the unit for tumor sample weight is labeled in grams and it is the same for all patients.
Comment 33:
“There is a lack of detailed statistical analysis or justification for the sample size (n=12) used in the study, which may affect the robustness and generalizability of the conclusions.”
Answer for Comment 33:
The participants were selected on a random basis during several months for the development of ex vivo analysis of NSCLC tumor samples. All the patients’ characteristics, procedures and experiments are presented and discussed in tables, figures, and legends to them. The statistical analysis for some data obtained in our work was shown in Figure 5.
Comment 34:
“The descriptions of Figures 1 and 2 lack clarity and precision. For example, it is not clear what specific cellular features are being highlighted or how the images support the conclusions drawn.”
Answer for Comment 34:
In the Results, in lines 300-307: “Figure 1. Representative images demonstrate isolation of cancer cells (a) from surgically resected tumor sample sq1 (weight 0.07 g), which is indicated by the red arrow in the Petri dish 5 cm in diameter, (b) in the cell suspension after separating fibrotic tissue in a sieve and (c-f) their analysis on (c, e) cell smears and (d, f) the ex vivo cell preparations after (c, d) Romanovsky–Giemsa staining and (e, f) the immunofluorescence assay with specific antibodies to different NSCLC and fibroblast markers (green and red signals). Nuclei are stained by DAPI (blue signal). (e, f) Colocalization of the markers is (e) yellow and (f) magenta (in the nuclei) signals on confocal immunofluorescent images. The scale bars are (c, d) 10, (e) 20, and (f) 5 mm.”; in Figure 1: Cancer-associated fibroblasts: CD10 (+) p40 (-) F-actin (+); Cancer cells of lung squamous cell carcinoma: PanCK (+) p40 (+)”
In the Results, in lines 325-329: “Figure 2. Representative images after Romanovsky–Giemsa staining demonstrate that the differentiation and specific features of the patients’ adenocarcinoma cells and their clusters can be defined not only (e-l) on the histological sections, but also (a-d) on the ex vivo cell preparations obtained from the same tumor samples. (e-h) Close-ups of the parts of the images (i-l). The scale bars are (i) 5, (a-d, j-l) 10, and (e-h) 50 mm.”; in Figure 2: Lung adenocarcinoma: Moderately differentiated, Poorly differentiated, Well-differentiated, Mucinous.”
Comment 35:
“The paper mentions specific markers such as TTF1, CK7, p40, and PanCK, but does not provide detailed methodology on how these markers were quantified or the statistical significance of their expression levels.”
Answer for Comment 35:
In the Results, in lines 361-362 (Table 2): “1 The number of the marker-positive cancer cells expressed as the percentage of the total number of the cancer cells examined…” Also, cancer cells with expression of lung cancer-specific markers on the ex vivo cell preparations and histological sections parallelly obtained from the same tumor samples were shown in Figures 1, 3A-C, and S2.
Comment 36:
“The study does not clearly state the use of control samples, making it difficult to determine the baseline against which the tumor samples were compared.”
Answer for Comment 36:
Now, in the Materials and Methods, in lines 213-216: “Actively growing human larynx carcinoma Hep-2 cells (courtesy of PhD. M.V. Solomatina of the Federal Research Center of Fundamental and Translational Medicine, Novosibirsk, Russia) were used as the internal control to validate the adequacy of the Ki-67 staining reaction…”
In the Results, in Figure 3A-C and S2, cancer cells both with the expression of lung cancer-specific markers and without the expression of them were shown.
Now, the Results, in lines 356-357 (Figure 3C): “Green arrows indicate the PD-L1-positive cancer cells, as solitary and in clusters.”
Now, in the Supplementary Materials, in lines 19-20 (Figure S2): “Red arrows indicate the CYP1A1-positive cancer cells, as solitary and in clusters.”
Comment 37:
“In Table 2, the presentation of data on immune cell composition is confusing, with percentages and raw numbers intermixed without clear explanation. This makes it difficult to interpret the data accurately.”
Answer for Comment 37:
In the Results, in lines 366-369 (Table 2): “4 Data are presented as the percentage of the number of immune cells of a particular type out of the total number of immune cells examined. 5 The total number of immune cells (all types) expressed as the percentage of the total number of the patients’ cells (cancer and immune) examined. (+), is present; (-), is absent;”
Comment 38:
“The discrepancy between ex vivo and histological analysis of PD-L1 expression is noted, but the reason for this discrepancy is not explored in detail, which is crucial for understanding the validity of the findings.”
Answer for Comment 38:
In the Discussion, in lines 494-503: “Notably, PD-L1 expression was detected only by ex vivo analysis in some NSCLC patients, while the assessment of the other cancer-specific and immune-related markers showed that the results obtained from the ex vivo cell preparations were absolutely comparable with those obtained from the histological sections of the same tumor and lung tissue samples for all the patients. Thus, in diagnosing PD-L1 that exhibit focal expression in tumor tissues [21-28], ex vivo analysis of NSCLC samples decreased the number of false negatives in the immunohistochemical examination, and so this analysis may be a method of choice in assessing the status of this parameter for cancer cells, especially those with low PD-L1 TPS, and treatment decision making.”
Also, in lines 573-580: “Secondly, our ex vivo analysis and conclusions concerning the expression of various markers, including PD-L1, and the immunologic landscape, are based on testing only one small part obtained from each surgically resected lung tumor, which is supposed to be highly heterogeneous within different NSCLC tumor areas. In this regard, a larger set of tumor samples from each NSCLC resection should be characterized by ex vivo analysis to contribute to a better understanding of cancer pathogenesis, including the immunological status, and the mechanisms leading to the modulation of PD-L1 expression.”
Comment 39:
“The paper claims that ex vivo analysis is more comprehensive than histological examination for measuring PD-L1 expression, but it does not provide sufficient evidence or rationale to support this claim.”
Answer for Comment 39:
In the Results, in lines 370-381: “Substantial variations in the results obtained from the ex vivo and histological analyses were observed only in the identification of cancer cells with PD-L1 membrane expression in some tumor samples. 10% and 2% of the PD-L1-positive cancer cells were found on the ex vivo cell preparations obtained from tumor samples sq1 and sq3, respectively, but not on the histological sections obtained from the same tumor samples and simultaneously stained in the immunofluorescence assay using clone ABM4E54 (Abcam) of mouse anti-PD-L1 antibodies (Table 2, Figure 2C). Of note, according to the medical records, PD-L1-positive cancer cells had not been identified in these tumors by a standard immunohistochemical analysis conducted in the Novosibirsk Regional Clinical Oncology Dispensary (Novosibirsk, Russia) using clone SP263 (Ventana) of rabbit anti-PD-L1 antibodies. For the other NSCLC patients, the results of testing PD-L1 were the same in all analyses and had a close identity with data in the medical records.”
In the Results, in lines 352-357 (Figure 3): “(C) PD-L1 expression is detected (u, w) in some lung squamous cell carcinoma cells only by ex vivo analysis… Green arrows indicate the PD-L1-positive cancer cells, as solitary and in clusters.”
In the Discussion, in lines 494-503: “Notably, PD-L1 expression was detected only by ex vivo analysis in some NSCLC patients, while the assessment of the other cancer-specific and immune-related markers showed that the results obtained from the ex vivo cell preparations were absolutely comparable with those obtained from the histological sections of the same tumor and lung tissue samples for all the patients. Thus, in diagnosing PD-L1 that exhibit focal expression in tumor tissues [21-28], ex vivo analysis of NSCLC samples decreased the number of false negatives in the immunohistochemical examination, and so this analysis may be a method of choice in assessing the status of this parameter for cancer cells, especially those with low PD-L1 TPS, and treatment decision making.”
Also, in lines 573-580: “Secondly, our ex vivo analysis and conclusions concerning the expression of various markers, including PD-L1, and the immunologic landscape, are based on testing only one small part obtained from each surgically resected lung tumor, which is supposed to be highly heterogeneous within different NSCLC tumor areas. In this regard, a larger set of tumor samples from each NSCLC resection should be characterized by ex vivo analysis to contribute to a better understanding of cancer pathogenesis, including the immunological status, and the mechanisms leading to the modulation of PD-L1 expression.”
Comment 40:
“There is no mention of data normalization or adjustment for potential confounding factors such as age, smoking status, or concomitant diseases, which could bias the results.”
Answer for Comment 40:
In the Discussion, in lines 569-580: “Although we have developed a method of ex vivo analysis to assess various characteristics of NSCLC and demonstrated some of its advantages over the routine histological assay, there are still several limitations that need to be addressed in future studies. First, the number of NSCLC samples tested in this work is small, and a larger number of tumor samples is needed to promote ex vivo analysis and validate the results. Secondly, our ex vivo analysis and conclusions concerning the expression of various markers, including PD-L1, and the immunologic landscape, are based on testing only one small part obtained from each surgically resected lung tumor, which is supposed to be highly heterogeneous within different NSCLC tumor areas. In this regard, a larger set of tumor samples from each NSCLC resection should be characterized by ex vivo analysis to contribute to a better understanding of cancer pathogenesis, including the immunological status, and the mechanisms leading to the modulation of PD-L1 expression.” In lines 591-592: “Finally, to evaluate the applicability and performance of ex vivo analysis in the NSCLC studies, further studies are required.”
Comment 41:
“The methodology section lacks details on the protocols used for cell preparation, staining, and analysis, which are crucial for reproducibility and validation of the findings.”
Answer for Comment 41:
In the Materials and Methods, in lines 166-259:
“2.2. Ex vivo isolation of cells and production of ex vivo cell preparations
Cells were isolated ex vivo from the samples of surgically resected tumor and attendant lung tissues with reliance on a technique that we had previously used to produce ex vivo cell cultures, mainly those of alveolar macrophages, from lung lesions surgically removed from patients with pulmonary tuberculosis [50]. In this work, samples were cut into small pieces and, to separate cell suspension containing cancer and immune cells from fibrotic tissue, were further rubbed through a metal screen of a sieve with pores 0.5-1.0 mm in diameter in phosphate-buffer saline (PBS, pH 7.4) without any enzymatic treatments. Cell pellets were centrifuged at 400 g for 5 minutes at room temperature. Precipitates were diluted in PBS and 100 µl of cell suspension was introduced into a single cytofunnel sample chamber (Tharmac Cellspin, Germany), which was linked to a glass single cytoslide (Tharmac Cellspin, Germany) for each ex vivo cell preparation obtained. Then, thin-layer cell preparations were made using the cytocentrifuge Awel MF20 (Domel, Slovenia) run at 400 g for 5 minutes at room temperature. The ex vivo cell preparations were immediately fixed with 4% formaldehyde solution in PBS for 10 minutes at room temperature. Six ex vivo cell preparations were made by the same method for each tumor sample and, in parallel, for each attendant lung tissue sample.
2.3. Cell staining
To visualize cancer cells and different types of immune cells, some ex vivo cell preparations were washed with PBS and stained with a mixture of azure, eosin, and methylene blue in Romanovsky–Giemsa stain (Minimed, Moscow, Russia) according to the manufacturer’s instruction.
After washing with PBS, the other ex vivo cell preparations were permeabilized with 0.3% Triton X-100 solution in PBS for 2 minutes, blocked in PBS solution containing 2% BSA, and incubated with the appropriate rabbit or mouse primary antibodies to human cancer cell markers: TTF1 (clone SP141, Abcam, England, ab227652), p40 (clone EPR17863-47, Abcam, England, ab203826), CK7 (clone PBM-12F1, PrimeBioMed, Moscow, Russia), PanCK, including the cytokeratin 5/6 (clone С11, PrimeBioMed, Moscow, Russia), Ki-67 (clone SP6, Thermo Fisher Scientific, USA, MA5-14520), AhR (clone RPT9, Abcam, England, ab2769), AhRR (Abcam, England, ab234817), CYP1A1 (MyBioSource, China, MBS178240); to immune-related markers: PD-L1 (clone ABM4E54, Abcam, England, ab210931), PD-1 (clone CAL20, Abcam, England, ab237728), CD14 (clone SP192, Spring Bioscience, USA, M492), IFNγ (clone 25718, Thermo Fisher Scientific, USA, MA5-23718), TNFα (Thermo Fisher Scientific, USA, P300A), FGFb (clone 6/basic FGF, BD Biosciences, USA, 610072), IL-1β (clone 2805, Thermo Fisher Scientific, USA, MA5-23691), IL-4 (Thermo Fisher Scientific, USA, PA5-25165), IL-10 (clone 945A2A5, Invitrogen, USA, AHC9102), IL-12 (clone 24945, Thermo Fisher Scientific, USA, MA5-23715), and to fibroblast markers: CD10 (clone GM-003, PrimeBioMed, Moscow, Russia), a membrane metalloendopeptidase [51], and TRITC-labeled phalloidin dye (Sigma-Aldrich, USA, P1951) to stain filamentous actin (F-actin), diluted 1:100 each.
For all ex vivo cell preparations, fluorescent visualization of bound primary antibodies was achieved using goat polyclonal secondary antibodies to rabbit IgG conjugated with DyLight 488 or DyLight 594 (Thermo Fisher Scientific, USA, 35553 or 35561, respectively) and to mouse IgG conjugated with Alexa 488 (Thermo Fisher Scientific, USA, A-11001) diluted 1:400 each. The ex vivo cell preparations were incubated with the appropriate antibodies for 60 minutes at room temperature. Fluorescent staining was analyzed using the ProLong Gold Antifade Mountant with DAPI (Thermo Fisher Scientific, USA, P36935). Actively growing human larynx carcinoma Hep-2 cells (courtesy of PhD. M.V. Solomatina of the Federal Research Center of Fundamental and Translational Medicine, Novosibirsk, Russia) were used as the internal control to validate the adequacy of the Ki-67 staining reaction and were stained with the appropriate antibodies and Alexa 488-labeled phalloidin dye (Thermo Fisher Scientific, USA, A12379) in parallel with the NSCLC preparations.
For some ex vivo preparations, biochemical visualization of bound primary antibodies was achieved using the Hydrogen Peroxidase Blocking Reagent with 3% H2O2 solution and the Prime Vision kit with horseradish peroxidase (HRP) and diaminobenzidine (DAB) (PrimeBioMed, Moscow, Russia) as a substrate according to the manufacturer’s instructions. After DAB staining, the cells were further counterstained with Romanovsky–Giemsa stain as described above.
2.4. Histology
For all the patients, the histological sections of the resected tumor and attendant lung tissues were prepared in parallel with ex vivo cell expansion. In brief, each resected specimen was cut into two. One portion of the pieces was collected for making ex vivo cell preparations as described above. The other portion was fixed with 4% formaldehyde solution in PBS for 15 hours at +4 °C. After fixation, the pieces of tumor and attendant lung tissue were washed with PBS, incubated with 30% sucrose in PBS (pH 7.4) for 20 hours at +4 °C, frozen in Tissue-Tek O.C.T. Compound (Sakura Finetek, USA, 4583) at -25 °C, and sectioned into 16-µm slices using a Microtome Cryostat HM550 (Microm, Germany) at the Shared Center for Microscopic Analysis of Biological Objects of the Institute of Cytology and Genetics, SB RAS (Novosibirsk, Russia). Sections were air-dried on SuperFrost Plus slides (Thermo Fisher Scientific, USA) and stained with Romanovsky–Giemsa stain or antibodies as described above in parallel with the ex vivo cell preparations.
All the histological sections (~ 4 x 4 mm each) were permeabilized with 0.3% Triton X-100 solution in PBS for 45 minutes, blocked, and incubated with the appropriate primary antibodies for 20 hours at +4 °C and with the appropriate secondary antibodies for 60 minutes at room temperature. Fluorescent staining was analyzed using the ProLong Gold Antifade Mountant with DAPI as described above.
For some histological sections, biochemical visualization of bound primary antibodies was achieved using the Hydrogen Peroxidase Blocking Reagent and the Prime Vision kit as described above. After DAB staining, the cells were further counterstained with Romanovsky–Giemsa stain as described above.
2.5. Microscopy
The ex vivo cell and histological preparations were examined at the Shared Center for Microscopic Analysis of Biological Objects of the Institute of Cytology and Genetics, SB RAS (Novosibirsk, Russia), using an Axioskop 2 plus microscope (Zeiss) and objectives with various magnifications (Zeiss), and photographed using an AxioCam HRc camera (Zeiss); the images were analyzed using the AxioVision 4.7 microscopy software (Zeiss). All the preparations stained with fluorescent dyes were examined under an LSM 780 laser scanning confocal microscope (Zeiss) using the ZEN 2010 software (Zeiss). All the ex vivo cell preparations were analyzed for cell composition and the expression of different markers for each cell type. The cancer and immune cells were counted separately on each ex vivo cell preparation for each patient in each test. All cancer cells (from 800 to 2000 cells) were analyzed in each ex vivo cell preparation. For the histological preparations, three serial tumor and attendant lung tissue sections were analyzed for each staining in each test.”
Comment 42:
“Comments on the Quality of English Language”
Answer for Comment 42:
I’m thankful to the Reviewer for this comment and feel sorry for the mistakes made during the Russian-English translation. Now, I corrected some language mistakes, but I rely on the help of the editors of Journal to correct other unseen language mistakes.
Once again, I would like to thank the Reviewer for careful reading of our manuscript and for their comments that led to improving of article’s contents.
Elena Ufimtseva,
PhD,
Research Institute of Biochemistry, Federal Research Center of Fundamental and Translational Medicine, Novosibirsk, Siberia, Russia.
